# A Square Peg in a Square Hole:
# Meta-Expert for Long-Tailed Semi-Supervised Learning

**Yaxin Hou** [1]   **Yuheng Jia** [1 2]

## Abstract

This paper studies the long-tailed semi-supervised learning (LTSSL) with distribution mismatch, where the class distribution of the labeled training data follows a long-tailed distribution and mismatches with that of the unlabeled training data. Most existing methods introduce auxiliary classifiers (experts) to model various unlabeled data distributions and produce pseudo-labels, but the expertises of various experts are not fully utilized. We observe that different experts are good at predicting different intervals of samples, e.g., long-tailed expert is skilled in samples located in the head interval and uniform expert excels in samples located in the medium interval. Therefore, we propose a dynamic expert assignment module that can estimate the class membership (i.e., head, medium, or tail class) of samples, and dynamically assigns suitable expert to each sample based on the estimated membership to produce high-quality pseudo-label in the training phase and produce prediction in the testing phase. We also theoretically reveal that integrating different experts' strengths will lead to a smaller generalization error bound. Moreover, we find that the deeper features are more biased toward the head class but with more discriminative ability, while the shallower features are less biased but also with less discriminative ability. We, therefore, propose a multi-depth feature fusion module to utilize different depth features to mitigate the model bias. Our method demonstrates its effectiveness through comprehensive experiments on the CIFAR-10-LT, STL-10-LT, and SVHN-LT datasets across various settings.

[1]School of Computer Science and Engineering, Southeast University, Nanjing, China. [2]Key Laboratory of New Generation Artificial Intelligence Technology and Its Interdisciplinary Applications (Southeast University), Ministry of Education, Nanjing, China. Correspondence to: Yuheng Jia <yhjia@seu.edu.cn>.

*Proceedings of the $42^{nd}$ International Conference on Machine Learning*, Vancouver, Canada. PMLR 267, 2025. Copyright 2025 by the author(s).

*Table 1.* Accuracy (%) of testing set under three different unlabeled data distributions with varying experts. In CPE (Ma et al., 2024), $E_2$ denotes uniform expert, while $E_1$ and $E_3$ denote long-tailed and inverse long-tailed experts, respectively. Our proposed method and Upper-E use the proposed dynamic expert assignment (DEA) module and the ground-truth class membership to select a specific expert, respectively. The dataset is CIFAR-10-LT with imbalance ratio $\gamma_l = 200$. † indicates our proposed method using the ground-truth class membership to select a specific expert.

| Distribution | Expert | Head | Medium | Tail | Overall |
|---|---|---|---|---|---|
| Consistent | $E_1$ | **94.67** | 74.10 | 38.73 | 69.66 |
| | $E_2$ | 87.23 | **77.30** | **71.60** | 78.57 |
| | $E_3$ | 3.57 | 72.35 | 68.47 | 50.55 |
| | Ours | 89.13 | 79.52 | 77.07 | **81.67** |
| | Upper-E | 94.67 | 77.30 | 68.47 | 79.86 |
| | Upper-E† | 94.17 | 77.53 | 85.93 | 85.04 |
| Uniform | $E_1$ | **93.47** | 74.73 | 66.83 | 77.98 |
| | $E_2$ | 86.93 | **77.60** | 87.83 | 83.47 |
| | $E_3$ | 0.53 | 74.55 | **90.20** | 57.04 |
| | Ours | 89.40 | 78.35 | 86.00 | **83.96** |
| | Upper-E | 93.47 | 77.60 | 90.20 | 86.14 |
| | Upper-E† | 93.40 | 78.40 | 90.43 | 86.51 |
| Inverse | $E_1$ | **93.56** | 74.25 | 75.87 | 80.53 |
| | $E_2$ | 83.90 | **78.58** | 92.67 | 84.40 |
| | $E_3$ | 57.00 | 77.43 | **95.10** | 76.60 |
| | Ours | 88.60 | 78.90 | 92.03 | **85.75** |
| | Upper-E | 93.56 | 78.58 | 95.10 | 88.03 |
| | Upper-E† | 92.20 | 80.05 | 96.40 | 88.60 |

## 1. Introduction

Over the last decade, extensive high-quality labeled data have improved the performance of deep neural networks (DNNs). However, in specialized domains such as medical diagnosis (Yuan et al., 2023; Zhang et al., 2023b), the scarcity and imbalance of labeled data can be a significant challenge due to the high costs associated with data collection or annotation (Chen et al., 2024). To solve this issue, semi-supervised learning (SSL) (Sohn et al., 2020; Berthelot et al., 2020; Zhang et al., 2021) has been proposed and become a popular method to utilize the easier and cheaper acquired unlabeled data to improve the performance of DNN models. Its core idea is generating pseudo-labels for unlabeled data and selecting high-confidence ones to train the model together with the labeled data, so as to obtain a better

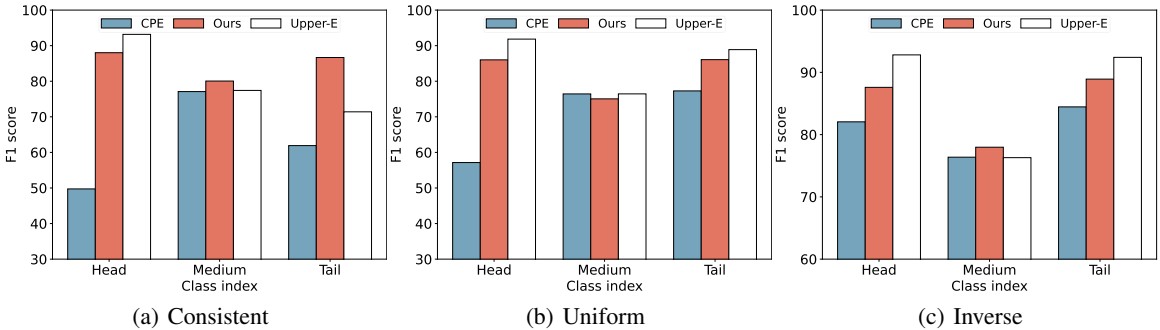

*Figure 1.* Comparison of F1 score (%) of pseudo-label predictions between CPE (Ma et al., 2024) and our proposed method under three different cases. Upper-E denotes the F1 score (%) of CPE with ground-truth class membership. The dataset is CIFAR-10-LT with imbalance ratio $\gamma_l = 200$. Our proposed method can generate pseudo-labels with higher F1 scores than CPE on all the cases, indicating its effectiveness in the utilization of unlabeled samples.

model than using labeled data only. However, traditional SSL usually assumes that the class distributions of the labeled and unlabeled data are balanced and consistent, which is easily violated in real-world applications. Specifically, data typically exhibit a long-tailed distribution, and the class distribution of unlabeled data is not always the same as that of the labeled data, i.e., unlabeled data may exhibit any one of the long-tailed, uniform, or inverse long-tailed distribution, which further exacerbates the difficulty of model training (Zhang et al., 2023c). This problem is known as long-tailed semi-supervised learning (LTSSL).

**A motivating example.** In the medical field, when collecting clinical data, we may obtain a long-tailed dataset from hospitals, i.e., many common disease cases (head classes) accompanied by very few rare disease cases (tail classes). However, the clinical data collected from a wide range of populations is unlabeled and characterized by an abundance of non-diseased individuals and a scarcity of diseased individuals, especially those with rare diseases. Thus, the unlabeled data distribution is mismatched with the labeled data distribution.

To mitigate the model bias arising from long-tailed distribution, long-tailed learning techniques, such as logit adjustment (Menon et al., 2021) and re-sampling (Xu et al., 2022), have been utilized in LTSSL to produce unbiased and high-quality pseudo-labels. Despite their effectiveness, they still cannot effectively solve the model bias resulting from distribution mismatch between labeled and unlabeled training data. Recently, ACR (Wei & Gan, 2023) proposes to handle the mismatched unlabeled data with various distributions by incorporating adaptive logit adjustment. While this method is effective, it relies on pseudo-labels generated by a single classifier (expert), limiting its performance. In response to this limitation, CPE (Ma et al., 2024) suggests training three classifiers (experts) to handle unlabeled data across various class distributions. However, CPE still suffers from the following two drawbacks. First, it employs three experts to

generate pseudo-labels simultaneously in the training phase, which may introduce more error pseudo-labels. Second, in the testing phase, it only employs the uniform expert for prediction, ignoring the different characteristics of three experts.

**Motivation**: We argue that each expert has its strength and weakness, e.g., the long-tailed expert is good at handling the samples in the head classes but not the samples in the medium and tail classes, while the uniform expert excels in medium classes but not in tail and head classes, and we should use different experts to process samples from different intervals, i.e., "a square peg in a square hole". To check this assumption, we first assume the class membership (i.e., head, medium, or tail class) of each sample is known. Then, we construct the Upper-E, which uses the long-tailed expert in CPE to produce pseudo-labels and predictions for head class samples in the training and testing phase, respectively, and the uniform and inverse long-tailed experts for medium and tail class samples. As shown in Fig. 1 and Table 1, this strategy can largely improve the quality of pseudo-labels in the training phase, and the prediction accuracy in the testing phase across all cases, indicating that each expert has its expertise. This observation motivates us to design a module to accurately estimate the class membership of each sample to realize the prior of "a square peg in a square hole".

To this end, in this paper, we propose a flexible and end-to-end LTSSL algorithm, namely Meta-Expert. To estimate the class membership of each sample, we propose a Dynamic Expert Assignment (DEA) module, which takes features from the encoder and logits from the three experts as input, to produce the probability (soft class membership) of assigning a specific expert to each sample. Subsequently, based on a multi-information fusion mechanism (Peng et al., 2023; 2022), we integrate the expertises of these three experts according to the estimated probabilities to construct an aggregator. The aggregator ensures the long-tailed expert dominates pseudo-label generation in the training phase and

*Table 2.* Comparison of accuracy (%) on testing set using three different depth features and the corresponding performance gap (%) between head and tail classes. We apply the K-means clustering on different depth features to produce the results, and we can clearly see that the feature is more biased towards the head class but more discriminative as the depth increases.

| Feature depth | Overall | Head | Medium | Tail | Gap |
|---|---|---|---|---|---|
| *Shallow* | 30.30 | 26.00 | 35.50 | 27.67 | 1.67 |
| *Middle* | 38.10 | 36.67 | 41.25 | 35.33 | 1.33 |
| *Deep* | 71.00 | 84.67 | 77.25 | 49.00 | 35.67 |

prediction in the testing phase when the sample belongs to the head class, and the uniform and inverse long-tailed experts dominate when the sample belongs to the medium and tail classes, respectively. As shown in Fig. 1, the proposed method can produce significantly higher-quality pseudo-labels than CPE in the training phase. And as shown in Table 1, the proposed method can produce significantly higher prediction accuracies than CPE (i.e., employing the uniform expert for prediction) in the testing phase. Note that the proposed Meta-Expert integrates the logits from the three experts in a soft manner, pushing different experts to learn better. Thus, in several cases, it even outperforms CPE with the ground-truth class membership in pseudo-label generation in the training phase (Upper-E in Fig. 1) and prediction in the testing phase (Upper-E in Table 1).

The proposed Meta-Expert can produce better pseudo-labels and predictions. More importantly, our theoretical analysis confirms that integrating different experts' expertises reduces the model's generalization error, thereby enhancing its overall performance. However, the model is still naturally biased towards the head classes due to the scarcity of tail class samples. Fortunately, as shown in Table 2, we observed that shallow features are relatively balanced although less discriminative, and deep features improve the discriminative ability but are less balanced. This phenomenon aligns with the known behavior of deep networks: shallow layers capture local patterns while deep layers learn global semantics. For long-tailed learning, since head and tail classes may share similar local patterns, shallow features exhibit balanced discriminability across classes. Meanwhile, deep layers predominantly encode head class semantics due to their overwhelming sample dominance, thus biasing predictions toward head classes. Motivated by this observation, we further propose a Multi-depth Feature Fusion (MFF) module to mitigate the model bias towards the head class by fusing features across different depths to achieve both balanced and discriminative representation, which also echoes the wisdom of the proverb, "a square peg in a square hole".

In summary, our **contributions** are as follows:

1. We demonstrate that the pseudo-label quality and prediction accuracy can be notably improved by incorporating the

expertises of different experts. Motivated by the empirical guide, we propose the Dynamic Expert Assignment (DEA) module to assign experts to different samples based on their specific expertises.

2. We further theoretically show that leveraging different experts' strengths efficiently will bring a lower generalization error bound.

3. We are the first to discover that shallow depth features are less biased than deep ones, and propose the Multi-depth Feature Fusion (MFF) module to help deal with the model bias towards the head class.

4. We reach the new state-of-the-art (SOTA) performances on the popular LTSSL benchmarks under various settings.

## 2. Related Work

**Semi-supervised learning** (SSL) uses both labeled and unlabeled training data to obtain a better model than using labeled training data only. Recent SSL methods are mostly based on consistency regularization, pseudo-labeling, or both. Consistency regularization methods (Miyato et al., 2019) are based on the manifold or smoothness assumption and apply consistency constraints to the final loss function. Pseudo-labeling methods (Chen et al., 2018) produce pseudo-labels for unlabeled training data according to the model's high-confidence predictions and then use them to assist the model training. As a representative method of combining both of these techniques, FixMatch (Sohn et al., 2020) encourages similar predictions between weak and strong augmentation views of an image, to improve model's performance and robustness. Afterward, many variants based on FixMatch have been proposed, such as FlexMatch (Zhang et al., 2021), FlatMatch (Huang et al., 2023), SoftMatch (Chen et al., 2023), FreeMatch (Wang et al., 2023), $(FL)^2$ (Lee et al., 2024), and WiseOpen (Yang et al., 2024). Despite the superior performance of the above methods, they cannot effectively handle the case where labeled data exhibit a long-tailed distribution.

**Long-tailed semi-supervised learning** (LTSSL) has gained increased attention due to its high relevance to real-world applications. It takes both the long-tailed distribution in long-tailed learning (LTL) and the limited labeled training data in SSL into consideration, which makes it more realistic and challenging. Existing LTSSL methods primarily improve the model performance by introducing LTL techniques (Li & Jia, 2025; Jia et al., 2024; Zhang et al., 2023a) to the off-the-shelf SSL methods like FixMatch (Sohn et al., 2020). For instance, ABC (Lee et al., 2021), CReST (Wei et al., 2021), BMB (Peng et al., 2025), and RECD (Park et al., 2024) sample more tail class samples to balance training bias towards the head class. SAW (Lai et al., 2022) introduces the class learning difficulty based weight to the consistency loss

to enhance the model's robustness, INPL (Yu et al., 2023) proposes to select unlabeled data by the in-distribution probability, CDMAD (Lee & Kim, 2024) proposes to refine pseudo-labels by the estimated classifier bias, CoSSL (Fan et al., 2022) introduces feature enhancement strategies to refine classifier learning, and ACR (Wei & Gan, 2023) proposes to incorporate adaptive logit adjustment to handle unlabeled training data across various class distributions. Very recently, CPE (Ma et al., 2024) proposes to train multiple classifiers (experts) to handle unlabeled data across various distributions and further enhances the pseudo-label quality through an ensemble strategy. Although effective, it lacks a comprehensive strategy to utilize the expertise of each expert in pseudo-label generation and unseen sample prediction, leading to sub-optimal performance.

More related literature and discussions are detailed in Appendix B.

## 3. Method

### 3.1. Problem Statement

In long-tailed semi-supervised learning (LTSSL), we have a labeled training dataset $\mathcal{D}_l = \{x_i^l, y_i^l\}_{i=1}^{N}$ of size $N$ and an unlabeled training dataset $\mathcal{D}_u = \{x_j^u\}_{j=1}^{M}$ of size $M$, where $\mathcal{D}_l$ and $\mathcal{D}_u$ share the same feature and label space, $x_j^u$ is the $j^{th}$ unlabeled sample, $x_i^l$ is the $i^{th}$ labeled sample with a ground-truth label $y_i^l \in \{1, \ldots, C\}$, and $C$ is the number of classes. Let $N_c$ denote the number of samples in the $c^{th}$ class of labeled dataset, we assume that the $C$ classes are sorted in descending order, i.e., $N_1 > N_2 > \cdots > N_c$, thus its imbalance ratio can be denoted as $\gamma_l = N_1/N_c$. As the label is inaccessible for unlabeled dataset, we denote the number of samples in its $c^{th}$ class by $M_c$, and define its imbalance ratio $\gamma_u$ in the same way as labeled dataset for theoretical illustration only. In this paper, we follow the previous works (Wei & Gan, 2023; Ma et al., 2024) to consider three cases of unlabeled data distribution, i.e., consistent, uniform, and inverse. Specifically, i) for consistent setting, we have $M_1 > M_2 > \cdots > M_c$ and $\gamma_u = \gamma_l$; ii) for uniform setting, we have $M_1 = M_2 = \cdots = M_c$ and $\gamma_u = 1$; iii) for inverse setting, we have $M_1 < M_2 < \cdots < M_c$ and $\gamma_u = 1/\gamma_l$. The goal of LTSSL is to learn a classifier $F : \mathbb{R}^d \longmapsto [1, \ldots, C]$ parameterized by $\theta$ on $\mathcal{D}_l$ and $\mathcal{D}_u$, that generalizes well on all classes.

### 3.2. Proposed Framework

**Multi-expert based LTSSL.** As the class distribution of the unlabeled training data may be inconsistent with that of the labeled ones, we follow the previous work (Ma et al., 2024) to train three experts to handle the unlabeled training data across various class distributions, i.e., long-tailed, uniform, and inverse long-tailed distributions. Specifically,

we attach two auxiliary classifiers on a typical SSL method like FixMatch (Sohn et al., 2020), and train each classifier (expert) with a specific logit adjustment intensity to realize that the first (long-tailed) expert is skilled in long-tailed distribution, and the second (uniform) and third (inverse long-tailed) experts are skilled in uniform and inverse ones, respectively. Similar to FixMatch, the loss $L_{base}$ for the base LTSSL includes a supervised classification loss on the labeled data and an unsupervised consistency regularization loss on the unlabeled data , i.e.,

$$
\begin{aligned}
L_{base} = &\sum_{k=1}^{Q} \frac{1}{B_l} \sum_{i=1}^{B_l} \ell\big(E_k\big(g(x_i^l)\big) + \tau_k \log \pi, \ y_i\big) \\
&+ \sum_{k=1}^{Q} \frac{1}{B_u} \sum_{j=1}^{B_u} \ell\big(E_k\big(g(x_j^u)\big), \ \hat{y}_{j,k}\big)\mathbb{I},
\end{aligned}
\tag{1}
$$

where $Q = 3$ denotes the number of classifiers (experts), $B_l$ and $B_u$ denote the batch sizes of labeled and unlabeled data, respectively, $\ell$ denotes the cross-entropy loss, $x_i^l$ and $x_j^u$ denote the $i^{th}$ labeled and $j^{th}$ unlabeled samples in the current batch, respectively, $g$ denotes the encoder, $\pi$ denotes the label frequency of the labeled data, $E_k$ denotes the expert trained by cross-entropy loss with a specific logit adjustment intensity $\tau_k$, $\hat{y}_{j,k}$ denotes the pseudo-label predicted by $E_k$ on the $j^{th}$ unlabeled sample in the current batch, and $\mathbb{I}$ denotes a binary sample mask to select samples with confidence larger than the threshold $t$. The first term in Eq. 1 is the supervised classification loss on the labeled data, and the second term defines the unsupervised consistency regularization loss on the unlabeled data.

**Dynamic expert assignment.** As shown in Fig. 1 and Table 1, we observe that each expert has its strength and weakness, i.e., long-tailed expert is skilled in handling head class samples but not medium and tail class samples, while the uniform and inverse long-tailed experts are skilled in handling medium and tail class samples, respectively. Based on this observation, we propose to gather the strengths of different experts. To this end, we first propose a dynamic expert assignment (DEA) module to estimate the class membership (i.e., head, medium, or tail class) of each sample.

As shown in Fig. 2, the DEA module adopts a multilayer perceptron (MLP) architecture and takes the feature from the encoder and logits from the three experts as input and outputs the soft class membership for each sample, i.e., $w = DEA([v, z_1, z_2, z_3])$, where $w = [w_1, w_2, w_3]$ denotes the probability of assigning each expert to produce pseudo-label and make prediction, $v$ and $z_k|_{k=1}^{3}$ denote the feature and logit generated by the encoder $g$ and expert $E_k|_{k=1}^{3}$ on the sample $x$, respectively. The parameters of the DEA module can be inferred by minimizing the following DEA loss $L_{dea}$,

$$
L_{dea} = \frac{1}{B_l} \sum_{i=1}^{B_l} \ell\big(w_i^l, \ s_i\big) + \frac{1}{B_u} \sum_{j=1}^{B_u} \ell\big(w_j^u, \ \hat{s}_j\big)\mathbb{I}, \tag{2}
$$

where $w_i^l$ and $w_j^u$ denote the probabilities of assigning each

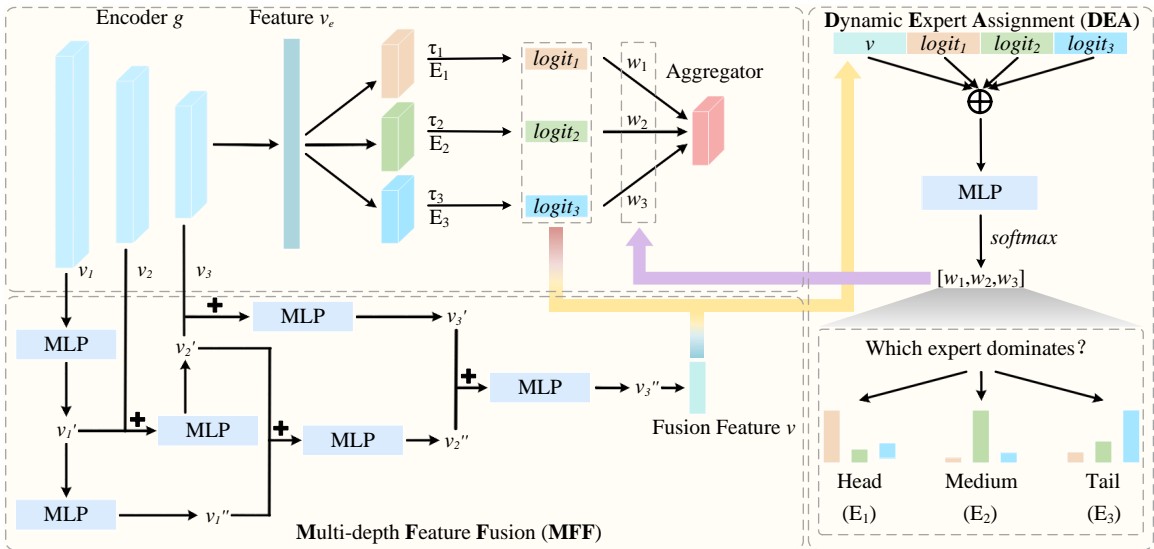

*Figure 2.* Overview of our Meta-Expert algorithm. Meta-Expert leverages a DEA module to adaptively assigns suitable expert to each sample to generate pseudo-label in the training phase and make prediction in the testing phase, crucial for integrating expertises of different experts and constructing the aggregator. $E_1$, $E_2$ and $E_3$ denote the long-tailed, uniform, and inverse long-tailed experts, respectively, $\tau_k$ denotes the logit adjustment intensity for expert $E_k$ ($k \in \{1, 2, 3\}$), and "✚" sign denotes adding different features.

expert to the $i^{th}$ labeled and $j^{th}$ unlabeled samples in the current batch, respectively, $s_i$ denotes the ground-truth class membership of the $i^{th}$ labeled sample in the current batch, and $\hat{s}_j$ denotes the pseudo class membership of the $j^{th}$ unlabeled sample in the current batch.

**Aggregator.** Subsequently, we construct an aggregator. The aggregator integrates the expertises of three experts through a weighted summation of their respective logits based on the estimated class membership $w$ by the DEA module, i.e.,

$$y_m = \sigma\Big(\sum_{k=1}^{Q} w_k z_k\Big), \tag{3}$$

where $y_m$ denotes the soft prediction produced by the aggregator and $\sigma(\cdot)$ denotes the softmax function. As the class membership estimated by the DEA module can reflect the class membership of each sample, aggregator in Eq. 3 ensures that the long-tailed expert dominates the pseudo-label generation in the training phase and prediction in the testing phase when the sample belongs to the head class, and the uniform and inverse long-tailed experts dominate when the sample belongs to the medium and tail classes, respectively.

Then, the META loss $L_{meta}$ for optimizing the overall network parameters based on the output of the aggregator is formulated as:

$$L_{meta} = \frac{1}{B_l}\sum_{i=1}^{B_l} \ell\big(y_{m,i}^l,\ y_i\big) + \frac{1}{B_u}\sum_{j=1}^{B_u} \ell\big(y_{m,j}^u,\ \hat{y}_j\big)\mathbb{I}, \tag{4}$$

where $y_{m,i}^l$ and $y_{m,j}^u$ denote the soft prediction produced by the aggregator on the $i^{th}$ labeled and $j^{th}$ unlabeled samples

in the current batch, respectively, $\hat{y}_j$ denotes the pseudo-label produced by the aggregator on the $j^{th}$ unlabeled sample in the current batch. As shown in Fig. 1 and Table 1, our proposed method can integrate the expertises of all experts and generate high-quality pseudo-labels and predictions in the training and testing phase, respectively.

**Multi-depth feature fusion.** Although the proposed method is effective in integrating the expertise of each expert, the model is still naturally biased towards the head class due to the scarcity of tail class samples. Fortunately, as shown in Table 2, we observe that different depth features have different bias intensities, i.e., shallow features are relatively balanced although less discriminative, and deep features are more discriminative and more biased. Such a phenomenon motivates us to use multiple depth features to learn a representation with a good trade-off between bias and discriminative ability.

Specifically, we propose a multi-depth feature fusion (MFF) module to fuse different depth features. As shown in Fig. 2, the MFF module involves several MLP layers and takes different depth features from the encoder as input and outputs the fusion feature $v$, i.e., $MFF(v_1, v_2, v_3) \longmapsto v$, where $v_1$, $v_2$ and $v_3$ denote the shallow, medium and deep features, respectively. Specifically, for shallow depth features $v_1$ and medium ones $v_2$, we first align their dimensions using an MLP and add them together. Subsequently, the resulting feature will be added to the subsequent depth's feature set to perform similar operations. Beside the addition operation, we can also concatenate different depth features in the MFF

module, which is investigated in Sec. 4.4. The parameters of the MFF module are updated by the end-to-end training.

### 3.3. Model Training and Prediction

In the training phase, we warm up the model by eighteen epochs with the base loss $L_{base}$ in Eq. 1. Then, we take the fusion feature $v$ from the MFF module and logit $z_k$ from the expert $E_k$ ($k \in \{1, 2, 3\}$) to calculate the DEA loss $L_{dea}$ in Eq. 2 and META loss $L_{meta}$ in Eq. 4, and together with the base loss $L_{base}$ to jointly optimize the model. The overall loss $L_{overall}$ is formulated as:

$$L_{overall} = L_{base} + L_{dea} + L_{meta}. \quad (5)$$

The whole framework of our method is shown in Fig. 2. The pseudo-code is summarized in Alg. 1. After training, we could obtain the predicted label of an unseen sample $x^*$ by selecting the label index with the highest confidence by $y_m$ in Eq. 3. Our code is made available[1].

### 3.4. Theoretical Analysis

We provide the generalization error bound for our Meta-Expert to analyze the factors that affect the model's generalization ability. Before providing the main results, we first define the true risk with respect to the classification model $f(x; \theta)$ as follows:

$$R(f) = \mathbb{E}_{(x,y)} \left[ \ell(f(x), y) \right]. \quad (6)$$

**Definition 1.** We aim to learn a good classification model by minimizing the empirical risk $\widehat{R}(f) = \widehat{R}_l(f) + \widehat{R}_u(f)$, where $\widehat{R}_l(f) = \frac{1}{N} \sum_{i=1}^{N} \ell(f(x_i), y_i)$ and $\widehat{R}_u(f) = \frac{1}{M} \sum_{j=1}^{M} \ell(f(x_j), y_j)$ denote the empirical risk on labeled and unlabeled training data, respectively. In SSL, we cannot minimize the empirical risk $\widehat{R}_u(f)$ directly, since the ground-truth label is inaccessible for unlabeled training data. Therefore, we need to train the model with $\widehat{R}'_u(f) = \frac{1}{\hat{M}} \sum_{j=1}^{\hat{M}} \ell(f(x_j), \hat{y}_j)$, where $\hat{M}$ denotes the number of selected high-confidence unlabeled samples and $\hat{y}_j$ denotes the pseudo-label of unlabeled sample $x_j$.

Let $\ell_{SSL} = \frac{1}{N} \sum_{i=1}^{N} \ell(f(x_i), y_i) + \frac{1}{\hat{M}} \sum_{j=1}^{\hat{M}} \ell(f(x_j), \hat{y}_j)$ be the loss for SSL, where $\hat{M}$ denotes the number of selected high-confidence unlabeled samples. Let the function space $\mathcal{H}_y$ for the label $y \in \{1, \ldots, C\}$ be $\{h : x \longmapsto f_y(x) | f \in \mathcal{F}\}$, where $f_y(x)$ denotes the predicted probability of the $y^{th}$ class. Let $\mathcal{R}_O(\mathcal{H}_y)$ be the expected Rademacher complexity (Mohri et al., 2018) of $\mathcal{H}_y$ with $O = N + M$ training samples (including $N$ labeled and $M$ unlabeled training samples). Let $\hat{f} = \operatorname{argmin}_{f \in \mathcal{F}} \widehat{R}(f)$ be the empirical risk minimizer, and $f^* = \operatorname{argmin}_{f \in \mathcal{F}} R(f)$ be the true risk minimizer. Then we have the following theorem.

[1]https://github.com/yaxinhou/Meta-Expert

---

**Algorithm 1** Training Process of the Proposed Method

1: **Input**: Labeled training dataset $\mathcal{D}_l$, unlabeled training dataset $\mathcal{D}_u$, hyper-parameter $t$.
2: **Output**: Encoder $g$, long-tailed expert $E_1$, uniform expert $E_2$, inverse long-tailed expert $E_3$, and parameters of DEA module $W_{dea}$ and MFF module $W_{mff}$.
3: Initialize the parameters of $g$, $E_1$, $E_2$, $E_3$, $W_{dea}$ and $W_{mff}$ randomly.
4: **for** epoch=1, 2, ... **do**
5:      **for** batch=1, 2, ... **do**
6:          Get expert $E_k$ prediction $z_k$ on a batch of data $B$;
7:          Calculate the base loss by Eq. 1;
8:          **if** warm up ended **then**
9:             Calculate the loss for DEA module by Eq. 2;
10:           Integrate the expertises of different experts by Eq. 3;
11:           Calculate the loss for optimizing the overall network parameters based on the integrated expertises by Eq. 4;
12:           Obtain the overall loss by Eq. 5;
13:          **end if**
14:          Update network parameters via gradient descent;
15:      **end for**
16: **end for**

---

**Theorem 1** (Generalization Error Bound). *Suppose that the loss function $\ell(f(x), y)$ is $\rho$-Lipschitz with respect to $f(x)$ for all $y \in \{1, \ldots, C\}$ and upper-bounded by $U$. Given the class membership $\eta \in \{1, \ldots, Q\}$ and overall pseudo-labeling error $\epsilon > 0$, if $\frac{1}{M} \sum_{j=1}^{M} \sum_{k=1}^{Q} \mathbb{I}(\eta_{j,k} = 1) |\mathbb{I}(max(f_k(x_j)) > t) - \mathbb{I}(\hat{y}_j = y_j)| \le \epsilon$, for any $\delta > 0$, with probability at least $1 - \delta$, we have:*

$$R(\hat{f}) - R(f^*) \le 2U\epsilon + 4\sqrt{2}\rho \sum_{y=1}^{C} \mathcal{R}_O(\mathcal{H}_y) + 2U\sqrt{\frac{\log \frac{2}{\delta}}{2O}}. \quad (7)$$

The proof of Theorem 1 is provided in Appendix D. It can be observed that the generalization performance of $\hat{f}$ mainly depends on two factors, i.e., the overall pseudo-labeling error $\epsilon$ and the number of training samples $O$. As $O \to \infty, \epsilon \to 0$, Theorem 1 shows that the empirical risk minimizer $\hat{f}$ will get closer to the true risk minimizer $f^*$. In CPE, the overall pseudo-labeling error is defined as $\epsilon_{CPE} = \frac{1}{Q^2} \sum_{i=1}^{Q} \sum_{j=1}^{Q} \epsilon_{i,j}$, where $\epsilon_{i,j}$ denotes the pseudo-labeling error of the $i^{th}$ expert on the unlabeled samples with class membership $\eta_j = 1$. While the counterpart of our Meta-Expert is defined as $\epsilon_{Ours} = \frac{1}{Q} \sum_{i=1}^{Q} \sum_{j=1}^{Q} \mathbb{I}_{i=j} \epsilon_{i,j}$, where $\mathbb{I}_{i=j}$ denotes a binary expert mask to assign each expert to select high-confidence unlabeled samples located in its skilled interval. As illustrated in Table 3, compared with CPE in the consistent case, with the estimated class membership by the DEA module, our

*Table 3.* Pseudo-labeling error rate (i.e., $\epsilon$) (%) and utilization ratio (i.e, $\hat{M}/M$) (%) under three different unlabeled data distributions with varying experts. In CPE (Ma et al., 2024), $E_2$ denotes uniform expert, while $E_1$ and $E_3$ denote long-tailed and inverse long-tailed experts, respectively. Our proposed method uses the DEA module to select a specific expert. The dataset is CIFAR-10-LT with imbalance ratio $\gamma_l = 200$.

| Distribution | Expert | Head | Medium | Tail | $\epsilon$ | $\hat{M}/M$ |
|---|---|---|---|---|---|---|
| | $E_1$ | 9.01 | 20.90 | 43.04 | 23.98 | 94.90 |
| | $E_2$ | 26.82 | 20.85 | 22.35 | 23.09 | 64.30 |
| Consistent | $E_3$ | 92.15 | 21.47 | 23.04 | 43.14 | 95.37 |
| | CPE | 42.66 | 21.07 | 29.48 | 30.07 | 84.86 |
| | Ours | 21.79 | 18.67 | 23.87 | **21.17** | **95.31** |
| | $E_1$ | 7.56 | 24.50 | 25.22 | 19.63 | 95.73 |
| | $E_2$ | 13.67 | 24.58 | 11.00 | 17.23 | 96.57 |
| Uniform | $E_3$ | 97.44 | 25.33 | 9.33 | 42.17 | 98.80 |
| | CPE | 39.56 | 24.81 | 15.19 | 26.34 | 97.03 |
| | Ours | 12.89 | 24.92 | 15.11 | **18.37** | **98.44** |
| | $E_1$ | 6.10 | 22.18 | 22.97 | 17.59 | 84.23 |
| | $E_2$ | 15.79 | 21.13 | 9.03 | 15.90 | 93.39 |
| Inverse | $E_3$ | 40.39 | 22.48 | 5.67 | 22.81 | 96.55 |
| | CPE | 20.76 | 21.93 | 12.56 | 18.77 | 91.39 |
| | Ours | 11.08 | 19.91 | 8.59 | **13.87** | **93.75** |

Meta-Expert achieves a smaller overall pseudo-labeling error (from 30.07 percentage points (pp) reduced to 21.17 pp) and higher unlabeled data utilization ratio (from 84.86 pp improved to 95.31 pp), and similar conclusions can be observed in uniform and inverse cases, which are beneficial for obtaining a smaller generalization error bound.

# 4. Experiments

## 4.1. Experimental Setting

**Dataset.** We perform our experiments on three widely-used datasets for the LTSSL task, including CIFAR-10-LT (Krizhevsky, 2009), SVHN-LT (Netzer et al., 2011), and STL-10-LT (Coates et al., 2011). We follow the dataset settings in ACR (Wei & Gan, 2023) and CPE (Ma et al., 2024), details are as below.

· *CIFAR-10-LT*: We test with four settings in the consistent case: $(N_1, M_1) = (1500, 3000)$ and $(N_1, M_1) = (500, 4000)$, with $\gamma \in \{150, 200\}$. In the uniform case, we test with $(N_1, M_1) = (1500, 300)$, with $\gamma_l \in \{150, 200\}$, and $\gamma_u$ being 1. In the inverse case, we test with $(N_1, M_c) = (1500, 3000)$, with $\gamma_l \in \{150, 200\}$, and $\gamma_u$ being $1/\gamma_l$.

· *SVHN-LT*: We test our method under $(N_1, M_1) = (1500, 3000)$ setting. The imbalance ratio $\gamma_l$ is set to 150 or 200. With a fixed $\gamma_l = 150$, we also test our method under $\gamma_u \in \{1, 1/150\}$ for the uniform and inverse cases.

· *STL-10-LT*: Since the ground-truth labels of unlabeled data in STL-10-LT are unknown, we conduct experiments by controlling the imbalance ratio of labeled data only. We set $N_1$ as 150 or 450, with $\gamma_l \in \{15, 20\}$, and directly use

the original unlabeled data.

**Baseline.** We compare our method with seven LTSSL algorithms published in top-conferences in the past few years, including SAW (Lai et al., 2022), Adsh (Guo & Li, 2022), DePL (Wang et al., 2022), ACR (Wei & Gan, 2023), BaCon (Feng et al., 2024), CPE (Ma et al., 2024), and SimPro (Du et al., 2024), which are all based on the typical SSL method FixMatch. For a fair comparison, we test these baselines and our Meta-Expert on the widely-used codebase USB[2]. We use the same dataset splits with no overlap between labeled and unlabeled training data for all datasets.

## 4.2. Implementation Details

We follow the default settings and hyper-parameters in USB, i.e., the batch size of labeled data $B_l$ is set to 64 and unlabeled data $B_u$ is set to 128, and the confidence threshold $t$ is set to 0.95. Moreover, we use the WRN-28-2 (Zagoruyko & Komodakis, 2016) architecture, and the SGD optimizer with learning rate 3e-2, momentum 0.9, and weight decay 5e-4 for training. We repeat each experiment over three different random seeds and report the mean performance and standard deviation. We conduct the experiments on a single GPU of NVIDIA A100 using PyTorch v2.3.1.

## 4.3. Main Result

**In the case of consistent distribution ($\gamma_l = \gamma_u$).** We initiate our investigation by conducting experiments in the scenario where $\gamma_l = \gamma_u$. The primary results for CIFAR-10-LT are presented in Table 4. It is clear that across all different training dataset sizes and imbalance ratios, Meta-Expert achieves higher classification accuracy than all the previous baselines on CIFAR-10-LT. For example, given $(N_1, M_1, \gamma) = (500, 4000, 200)$, Meta-Expert surpasses all previous baselines by up to 2.47 pp. When moving to SVHN-LT dataset in Table 5, Meta-Expert performs comparable to the previous SOTA method BaCon, surpassing other baselines by up to 0.66 pp given $(N_1, M_1, \gamma) = (1500, 3000, 200)$.

**In the case of mismatched distribution ($\gamma_l \neq \gamma_u$).** In practical applications, the distribution of unlabeled data might significantly differ from that of labeled data. Therefore, we investigate uniform and inverse class distributions, such as setting $\gamma_u$ to 1 or 1/200 for CIFAR-10-LT. For STL-10-LT dataset, as the ground-truth labels of the unlabeled data are unknown, we only control the imbalance ratio of the labeled data. The results are presented in Tables 4 and 5, where we can find Meta-Expert can

[2]https://github.com/microsoft/Semi-supervised-learning

*Table 4.* Comparison of accuracy (%) on CIFAR-10-LT under $\gamma_l = \gamma_u$ and $\gamma_l \neq \gamma_u$ settings. We set $\gamma_l$ to 150 and 200 for CIFAR-10-LT. We use **bold** to mark the best results.

| Algorithm | CIFAR-10-LT | | | | CIFAR-10-LT | | | |
|---|---|---|---|---|---|---|---|---|
| | $N_1 = 500, M_1 = 4000$ | | $N_1 = 1500, M_1 = 3000$ | | $N_1 = 1500, M_c = 3000$ | | $N_1 = 1500, M_c = 3000$ | |
| | $\gamma_l = 200$ $\gamma_u = 200$ | $\gamma_l = 150$ $\gamma_u = 150$ | $\gamma_l = 200$ $\gamma_u = 200$ | $\gamma_l = 150$ $\gamma_u = 150$ | $\gamma_l = 200$ $\gamma_u = 1$ | $\gamma_l = 200$ $\gamma_u = 1/200$ | $\gamma_l = 150$ $\gamma_u = 1$ | $\gamma_l = 150$ $\gamma_u = 1/150$ |
| Supervised | 41.15 ±1.15 | 43.88 ±1.61 | 56.83 ±1.10 | 59.82 ±0.32 | 56.83 ±1.10 | 56.83 ±1.10 | 59.82 ±0.32 | 59.82 ±0.32 |
| FixMatch (NIPS 20) | 61.74 ±0.81 | 65.68 ±0.67 | 69.39 ±0.56 | 72.15 ±0.94 | 65.59 ±0.18 | 63.98 ±0.36 | 69.07 ±0.74 | 65.24 ±0.63 |
| w / SAW (ICML 22) | 61.22 ±4.11 | 68.51 ±0.16 | 74.15 ±1.52 | 77.67 ±0.14 | 78.60 ±0.23 | 70.55 ±0.48 | 80.02 ±0.50 | 73.67 ±0.50 |
| w / Adsh (ICML 22) | 62.04 ±1.31 | 66.55 ±2.94 | 67.13 ±0.39 | 73.96 ±0.47 | 71.06 ±0.77 | 65.68 ±0.44 | 73.65 ±0.36 | 66.51 ±0.69 |
| w / DePL (CVPR 22) | 69.21 ±0.62 | 71.95 ±2.54 | 73.23 ±0.62 | 76.58 ±0.12 | 73.26 ±0.46 | 69.35 ±0.26 | 75.62 ±0.86 | 71.23 ±0.54 |
| w / ACR (CVPR 23) | 71.92 ±0.94 | 76.72 ±1.13 | 79.96 ±0.85 | 81.81 ±0.49 | 81.18 ±0.73 | 81.23 ±0.59 | 83.46 ±0.42 | 84.63 ±0.66 |
| w / BaCon (AAAI 24) | 66.41 ±0.31 | 71.33 ±1.75 | 78.64 ±0.40 | 81.63 ±0.44 | 77.89 ±0.97 | 81.87 ±0.16 | 82.05 ±1.09 | 83.30 ±1.12 |
| w / CPE (AAAI 24) | 67.45 ±1.27 | 76.77 ±0.53 | 78.12 ±0.34 | 82.25 ±0.34 | 83.46 ±0.03 | 84.07 ±0.40 | 84.50 ±0.40 | 85.52 ±0.02 |
| w / SimPro (ICML 24) | 59.94 ±0.73 | 65.54 ±3.17 | 75.37 ±0.74 | 77.18 ±0.38 | 73.05 ±0.35 | 75.33 ±2.85 | 76.12 ±1.11 | 79.42 ±2.78 |
| w / Meta-Expert (Ours) | **74.39** ±0.46 | **77.19** ±0.58 | **80.63** ±0.83 | **82.52** ±0.40 | **83.90** ±0.11 | **85.71** ±0.03 | **84.91** ±0.14 | **86.78** ±0.31 |

*Table 5.* Comparison of accuracy (%) on STL-10-LT and SVHN-LT under $\gamma_l = \gamma_u$ and $\gamma_l \neq \gamma_u$ settings. We set $\gamma_l$ to 15 and 20 for STL-10-LT, and $\gamma_l$ to 150 and 200 for SVHN-LT. We use **bold** to mark the best results. $N/A$ denotes the unknown $\gamma_u$ in STL-10-LT since its inaccessible ground-truth label for unlabeled dataset.

| Algorithm | STL-10-LT | | | | SVHN-LT | | | |
|---|---|---|---|---|---|---|---|---|
| | $N_1 = 150, M_1 = 100k$ | | $N_1 = 450, M_1 = 100k$ | | $N_1 = 1500, M_1 = 3000$ | | $N_1 = 1500, M_c = 3000$ | |
| | $\gamma_l = 20$ $\gamma_u = N/A$ | $\gamma_l = 15$ $\gamma_u = N/A$ | $\gamma_l = 20$ $\gamma_u = N/A$ | $\gamma_l = 15$ $\gamma_u = N/A$ | $\gamma_l = 200$ $\gamma_u = 200$ | $\gamma_l = 150$ $\gamma_u = 150$ | $\gamma_l = 150$ $\gamma_u = 1$ | $\gamma_l = 150$ $\gamma_u = 1/150$ |
| Supervised | 40.44 ±1.42 | 42.31 ±0.42 | 56.56 ±1.07 | 59.81 ±0.45 | 84.10 ±0.05 | 86.14 ±0.50 | 86.14 ±0.50 | 86.14 ±0.50 |
| FixMatch (NIPS 20) | 56.12 ±1.38 | 60.63 ±0.92 | 68.33 ±0.80 | 71.55 ±0.74 | 91.36 ±0.15 | 91.99 ±0.18 | 93.94 ±0.79 | 90.25 ±2.45 |
| w / SAW (ICML 22) | 66.62 ±0.34 | 67.00 ±0.79 | 74.59 ±0.13 | 75.58 ±0.28 | 92.17 ±0.10 | 92.27 ±0.01 | 94.41 ±0.38 | 91.42 ±0.41 |
| w / Adsh (ICML 22) | 66.56 ±0.61 | 66.72 ±0.32 | 72.95 ±0.45 | 74.28 ±0.24 | 90.87 ±0.32 | 91.68 ±0.25 | 94.04 ±0.68 | 88.71 ±0.52 |
| w / DePL (CVPR 22) | 66.10 ±0.63 | 67.02 ±0.89 | 73.43 ±0.11 | 74.55 ±0.14 | 92.16 ±0.16 | 92.85 ±0.04 | 94.12 ±0.63 | 87.86 ±0.50 |
| w / ACR (CVPR 23) | 69.24 ±0.95 | 68.74 ±0.95 | 78.13 ±0.29 | 78.55 ±0.50 | 92.90 ±0.40 | 93.52 ±0.32 | 91.11 ±0.17 | 92.03 ±0.34 |
| w / BaCon (AAAI 24) | 66.68 ±0.38 | 68.26 ±1.16 | 77.29 ±0.23 | 77.73 ±0.40 | 93.30 ±0.15 | 93.94 ±0.21 | 94.54 ±0.42 | 93.69 ±0.41 |
| w / CPE (AAAI 24) | 68.01 ±0.65 | 67.07 ±1.72 | 78.02 ±0.14 | 78.71 ±0.24 | 85.79 ±0.54 | 86.31 ±0.05 | 94.14 ±0.24 | 93.06 ±0.34 |
| w / SimPro (ICML 24) | 43.65 ±0.55 | 44.45 ±0.98 | 57.23 ±1.43 | 60.33 ±0.59 | 92.51 ±0.71 | 93.94 ±0.10 | 94.59 ±0.28 | **94.76** ±0.41 |
| w / Meta-Expert (Ours) | **71.19** ±0.07 | **69.23** ±0.82 | **80.18** ±1.21 | **79.98** ±0.33 | **93.56** ±0.09 | **93.99** ±0.07 | **94.66** ±0.23 | 94.24 ±0.19 |

consistently and significantly outperform baseline algorithms on CIFAR-10-LT, validating its effectiveness to cope with varying class distributions of unlabeled data. Concretely, Meta-Expert surpasses the previous SOTA method by 1.64 pp and other baselines by up to 4.48 pp with $(N_1, M_1, \gamma_l, \gamma_u) = (1500, 3000, 200, 1/200)$. For STL-10-LT dataset, Meta-Expert surpasses the previous SOTA method by 1.27 pp and other baselines by up to 1.43 pp with $N_1 = 450$ and $\gamma_l = 15$. On SVHN-LT dataset, Meta-Expert achieves comparable performances with SimPro, surpassing other baselines by up to 0.55 pp.

In summary, our method outperforms almost all previous baselines regardless of training dataset sizes, imbalance ratios, and unlabeled training data distributions. We also evaluate all methods on FreeMatch (Wang et al., 2023) in Appendix C, where we can get a similar conclusion.

### 4.4. Ablation Study

**The effect of each module.** In Table 6, we evaluate the contribution of each key component in Meta-Expert. Specifically, we set $N_1$ to 1500 and $M_1$ to 3000, and perform experiments on CIFAR-10-LT and SVHN-LT. According to Table 6, we can observe that both DEA and MFF bring significant improvements. For example, on CIFAR-10-LT with $\gamma_l = \gamma_u = 200$, DEA and MFF bring accuracy gains of 0.65 pp and 1.26 pp, respectively, and the accuracy is improved up to 3.10 pp when using them together. When evaluating the overall performance, the DEA module provides an average 1.68 pp accuracy improvement across all imbalance ratios, showing relatively greater performance gains; while the MFF module delivers a 0.76 pp average gain, which though comparatively smaller, remains statistically significant. The combined DEA+MFF configuration achieves 2.42 pp improvement, confirming their complementary effectiveness and synergistic interaction. These improvements conclusively validate the effectiveness of our proposed modules in enhancing model robustness across diverse imbalance ratios.

Note that MFF improves the accuracy of all cases except the inverse case ($\gamma_u = 1/200$). This phenomenon can be attributed to two reasons. First, the overall imbalance ratio will be reduced as the inverse long-tailed unlabeled data ($\gamma_u = 1/200$) complements the long-tailed labeled data

*Table 6.* Comparison of accuracy (%) on with and without the DEA and MFF modules. The datasets are CIFAR-10-LT with $(N_1, M_1, \gamma_l) = (1500, 3000, 200)$ and SVHN-LT with $(N_1, M_1, \gamma_l) = (1500, 3000, 150)$.

| CPE | | CIFAR-10-LT($\gamma_u$) | | | SVHN-LT($\gamma_u$) | | |
|---|---|---|---|---|---|---|---|
| w/ DEA | w/ MFF | 200 | 1 | 1/200 | 150 | 1 | 1/150 |
| | | 78.57 | 83.47 | 84.40 | 86.26 | 93.82 | 92.62 |
| ✓ | | 79.22 | 83.61 | 84.58 | **93.90** | 94.30 | 93.60 |
| | ✓ | 79.83 | 83.89 | 83.10 | 90.49 | 92.89 | 93.47 |
| ✓ | ✓ | **81.67** | **83.96** | **85.75** | 93.89 | **94.34** | **94.05** |

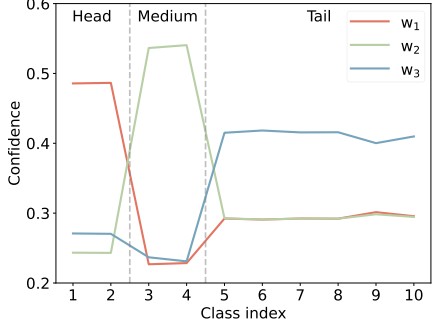

*Figure 3.* Confidence level for assigning a specific expert to the head, medium, and tail class samples. The datasets is CIFAR-10-LT with $(N_1, M_1, \gamma_l) = (1500, 3000, 200)$. We follow the setting in CPE to consider the first two of all classes as head classes, the last six as tail classes, and the remaining classes as the medium. We clearly see that when a sample belongs to the head class, $w_1$ is the largest, and $w_2$ and $w_3$ are the largest when a sample belongs to the medium and tail classes, respectively.

$(\gamma_l = 200)$. Second, without the DEA model, our method uses all experts to generate pseudo-labels in the training phase, which introduces more error pseudo-labels and limits the MFF module's effectiveness. When using the two modules (DEA+MFF) together, the DEA module reduces the quantity of error pseudo-labels, thus, the effectiveness of the MFF module is released completely.

**Feature combination strategy.** We evaluate the performance of using different feature combination strategies in the MFF module. As shown in Table 7, we observe that the addition operation outperforms concatenation, achieving a 0.58 pp ∼ 1.35 pp higher accuracy. Based on these empirical results, we adopt the addition operation for fusing multi-depth features.

**How does the DEA module improve the performance?** As previously mentioned, DEA can correctly estimate the class membership of given samples, which is essential to integrate the expertise of each expert and realize "a square peg in a square hole". To validate this assumption, we plot the soft class membership estimated by DEA in Fig. 3. As can be seen, DEA precisely assigns the head expert to the head class samples, and medium and tail experts to the medium and tail class samples, respectively. Based on this

*Table 7.* Comparison of accuracy (%) on utilizing different feature combination strategies in the MFF module. *add* and *con* denote using the addition and concatenation feature operation strategies, respectively. The dataset is CIFAR-10-LT with $(N_1, M_1, \gamma_l) = (1500, 3000, 200)$.

| Strategy | $\gamma_u$ | | |
|---|---|---|---|
| | 200 | 1 | 1/200 |
| *con* | 80.32 | 83.38 | 84.65 |
| *add* | **81.67** | **83.96** | **85.75** |

accurate estimation, our method can produce higher-quality pseudo-labels in the training phase, as can be seen in Fig. 1. Simultaneously, according to Table 1, the accuracies of our method in testing phase are also the highest in all cases. The above observations verify that our method utilizes the expertises of all three experts effectively.

**Computational overhead analysis.** While our proposed modules (three experts, MFF, and DEA) introduce a controlled parameter increase of 13.3 pp (1.5M → 1.7M), this design achieves a strategically balanced efficiency-performance trade-off. Experimental results on CIFAR-10-LT demonstrate: a 6.4 pp increase in epoch time (234.5s → 249.5s), a 1.6s increase in inference time for evaluating 10,000 samples (7.1s → 8.8s), and a substantial accuracy improvement of 3.5 pp (71.9 pp → 74.4 pp). These results collectively indicate significant performance enhancement with modest computational overhead.

## 5. Conclusion

In this work, we address the LTSSL problem from a fresh perspective, i.e., automatically integrating the expertises of various experts to produce high-quality pseudo-labels and predictions. We also theoretically prove that effectively exploiting different experts' expertises can reduce the generalization error bound. Specifically, the proposed Meta-Expert algorithm comprises a dynamic expert assignment module and a multi-depth feature fusion module, the former can assign a suitable expert to each sample based on the estimated class membership, and the latter relieves the biased training by fusing different depth features based on the observation that deep feature is more biased towards the head class but more discriminative, which has not been observed before. We validate our proposed method's effectiveness through extensive experiments across three widely-used LTSSL datasets with different imbalance ratios and unlabeled data distributions, reaching the new state-of-the-art performance. The ablation results show that both the DEA and MFF modules contribute to the performance improvement. Moreover, the DEA module can correctly estimate the class membership of given samples, which is essential to integrate the expertise of each expert.

## Acknowledgements

This work was supported by the National Natural Science Foundation of China under Grant U24A20322. This research work is also supported by the Big Data Computing Center of Southeast University.

## Impact Statement

This paper presents work whose goal is to advance the field of Machine Learning. Specifically, we propose a new long-tailed semi-supervised learning algorithm to improve the performance in distribution mismatched scenarios by integrating the expertises of various experts to produce high-quality pseudo-labels and predictions. We also theoretically prove that the model's generalization error can be reduced by integrating the expertises of different experts. There are many potential societal consequences of our work, none which we feel must be specifically highlighted here.

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

# Appendix

## A. overview of the Appendix

Our appendix consists of three main sections:

- More related literature and discussions.

- Additional evaluation on FreeMatch (Wang et al., 2023).

- Proof of the theorem in Sec. 3.4: theoretical analysis, i.e., Theorem 1 (generalization error bound).

## B. More Related Literature and Discussions

### B.1. Long-tailed learning (LTL)

Long-tailed learning (LTL) is tailored for the long-tailed distribution exhibiting in real-world applications (Pan et al., 2023), which aims to improve the performance of the tail class without compromising that of the head class. The existing methods can be roughly grouped into three categories: re-sampling, logit adjustment, and ensemble learning. Re-sampling (Bai et al., 2023; Liu et al., 2022; Xu et al., 2022) adjusts the number of samples for each class, i.e., under-sampling the head class or over-sampling the tail class. Logit adjustment (Cao et al., 2019; Menon et al., 2021; Kini et al., 2021) seeks to resolve the class imbalance by adjusting the predicted logit of the classifier. Ensemble learning (Aimar et al., 2023; Du & Wu, 2023; Zhang et al., 2022) based methods combine multiple classifiers (experts) to improve the performance and robustness of the model. While these methods have made significant progress in long-tailed learning, they often fail to achieve the expected performance gains when directly applied to long-tailed semi-supervised learning (LTSSL), particularly when the distributions are mismatched between labeled and unlabeled training data.

### B.2. Connections and Differences Compared with Multiple Experts Based Methods

In the fields of long-tailed learning (LTL) and long-tailed semi-supervised learning (LTSSL), several works have employed multiple experts to enhance the model's performance. Nevertheless, they are significantly different from the proposed Meta-Expert.

In LTSSL, only CPE (Ma et al., 2024) utilizes multiple experts to handle unlabeled data across various class distributions. CPE, however, lacks integrating the multiple experts, it only employs three experts to generate pseudo-label simultaneously in the training phase and the uniform expert to make prediction in the testing phase. Consequently, CPE may introduce more error pseudo-labels, thereby limiting its performance.

In contrast to LTSSL, numerous studies incorporate multiple experts in LTL, such as RIDE (Wang et al., 2021), SADE (Zhang et al., 2022), and BalPoE (Aimar et al., 2023). All of these methods are designed for supervised learning and are incapable of handling semi-supervised learning (SSL). Extending them to SSL requires extra effort, as exploiting unlabeled samples is not trivial.

RIDE seeks to learn from multiple independent and diverse experts, making predictions through the averaging of their outputs. The router in RIDE primarily focuses on minimizing the number of experts to reduce the computational cost during the testing phase. SADE leverages self-supervision on testing set to aggregate the learned multiple experts and subsequently utilize the aggregated ones to make prediction in the testing phase. Although effective, it lacks efforts to guide different experts to learn better in their expertises during the training phase. BalPoE refines the logit adjustment intensity across all three experts and averages their outputs to minimize the balanced error while ensuring Fisher consistency.

Different from the above works, we find that different experts excel at predicting different intervals of samples, e.g., a long-tailed/uniform/inverse long-tailed expert is skilled in samples located in the head/medium/tail interval. Consequently, we propose the dynamic expert assignment (DEA) module to estimate the class membership of samples and dynamically assign suitable experts to each sample based on the estimated membership to produce high-quality pseudo-labels in the training phase (Fig. 1) and excellent predictions in the testing phase (Tables 4, 5, 8, and 9). Moreover, Fig. 3 proves the class membership estimated by the DEA module is accurate.

Finally, for the first time, we observe that shallow features are relatively balanced although less discriminative, and deep features improve the discriminative ability but are less balanced, thus proposing the multi-depth feature fusion (MFF) module to make the model both discriminative and balanced (Table 2).

In summary, the differences between our Meta-Expert and existing methods are substantial, even though they also utilize a multi-expert network architecture.

## C. Evaluation on FreeMatch

To further illustrate our method's universality, we reproduce our method and all the compared methods by taking FreeMatch (Wang et al., 2023) as the base SSL method and evaluate them on CIFAR-10-LT, SVHN-LT, and STL-10-LT with the same settings used in the main paper. Tables 8 and 9 present the main results, indicating that our Meta-Expert can produce higher prediction accuracies by incorporating the expertises of different experts. For example, on CIFAR-10-LT, given $(N_1, M_1, \gamma) = (500, 4000, 200)$, Meta-Expert surpasses all previous baselines by up to 7.05 pp in the case of $\gamma_l = \gamma_u$. Moreover, in the case of $\gamma_l \neq \gamma_u$, Meta-Expert surpasses the previous SOTA method by 1.02 pp and outperforms all other baselines by 1.90 pp given $(N_1, M_1, \gamma_l, \gamma_u) = (1500, 3000, 150, 1)$. When moving to SVHN-LT dataset, Meta-Expert surpasses the previous SOTA method by 0.55 pp and other baselines by up to 3.29 pp given $(N_1, M_1, \gamma_l, \gamma_u) = (1500, 3000, 150, 1/150)$. On STL-10-LT, Meta-Expert performs comparable to the previous SOTA method SimPro, surpassing other baselines by up to 0.62 pp. All of these results further illustrate our method's effectiveness.

*Table 8.* Comparison of accuracy (%) on CIFAR-10-LT under $\gamma_l = \gamma_u$ and $\gamma_l \neq \gamma_u$ settings. We set $\gamma_l$ to 150 and 200 for CIFAR-10-LT. We use **bold** to mark the best results.

| Algorithm | CIFAR-10-LT | | | | CIFAR-10-LT | | | |
|---|---|---|---|---|---|---|---|---|
| | $N_1 = 500, M_1 = 4000$ | | $N_1 = 1500, M_1 = 3000$ | | $N_1 = 1500, M_c = 3000$ | | $N_1 = 1500, M_c = 3000$ | |
| | $\gamma_l = 200$ $\gamma_u = 200$ | $\gamma_l = 150$ $\gamma_u = 150$ | $\gamma_l = 200$ $\gamma_u = 200$ | $\gamma_l = 150$ $\gamma_u = 150$ | $\gamma_l = 200$ $\gamma_u = 1$ | $\gamma_l = 200$ $\gamma_u = 1/200$ | $\gamma_l = 150$ $\gamma_u = 1$ | $\gamma_l = 150$ $\gamma_u = 1/150$ |
| Supervised | 41.15 ±1.15 | 43.88 ±1.61 | 56.83 ±1.10 | 59.82 ±0.32 | 56.83 ±1.10 | 56.83 ±1.10 | 59.82 ±0.32 | 59.82 ±0.32 |
| FreeMatch (ICLR 23) | 63.35 ±0.49 | 68.03 ±0.68 | 69.83 ±1.36 | 73.00 ±0.63 | 78.99 ±0.53 | 71.33 ±0.36 | 79.60 ±0.72 | 72.76 ±0.79 |
| w / SAW (ICML 22) | 59.31 ±1.26 | 65.69 ±0.94 | 72.05 ±0.87 | 74.69 ±0.85 | 80.16 ±0.25 | 73.24 ±0.57 | 81.90 ±1.00 | 73.45 ±0.24 |
| w / Adsh (ICML 22) | 62.64 ±1.11 | 66.22 ±1.54 | 69.27 ±1.54 | 73.81 ±0.53 | 71.88 ±0.47 | 65.16 ±0.06 | 73.14 ±0.57 | 66.29 ±0.76 |
| w / DePL (CVPR 22) | 65.04 ±1.07 | 69.65 ±1.43 | 70.66 ±0.98 | 73.29 ±0.53 | 80.37 ±0.49 | 72.28 ±0.17 | 80.14 ±1.03 | 73.20 ±1.37 |
| w / ACR (CVPR 23) | 58.36 ±2.35 | 60.98 ±1.24 | 70.24 ±1.99 | 72.55 ±1.34 | 81.89 ±0.04 | 82.68 ±0.21 | 83.49 ±0.36 | 83.85 ±0.41 |
| w / BaCon (AAAI 24) | 68.63 ±0.77 | 72.69 ±0.68 | 75.97 ±1.34 | 77.71 ±1.36 | 83.20 ±0.11 | 82.48 ±0.08 | 83.91 ±0.42 | 83.66 ±1.25 |
| w / CPE (AAAI 24) | 66.82 ±1.36 | 69.80 ±1.28 | 77.28 ±0.63 | 79.24 ±0.30 | 83.59 ±0.17 | 80.37 ±1.11 | 84.37 ±0.14 | 79.12 ±0.80 |
| w / SimPro (ICML 24) | 64.20 ±0.39 | 69.17 ±3.31 | 78.57 ±0.72 | 80.49 ±0.39 | 80.82 ±0.24 | 77.59 ±0.72 | 81.30 ±0.98 | 79.67 ±0.99 |
| w / Meta-Expert (Ours) | **75.68** ±0.28 | **78.26** ±0.63 | **80.53** ±0.71 | **82.69** ±0.51 | **84.30** ±0.29 | **85.32** ±0.40 | **85.39** ±0.48 | **85.89** ±0.79 |

*Table 9.* Comparison of accuracy (%) on STL-10-LT and SVHN-LT under $\gamma_l = \gamma_u$ and $\gamma_l \neq \gamma_u$ settings. We set $\gamma_l$ to 15 and 20 for STL-10-LT, and $\gamma_l$ to 150 and 200 for SVHN-LT. We use **bold** to mark the best results. $N/A$ denotes the unknown $\gamma_u$ in STL-10-LT since its inaccessible ground-truth label for unlabeled dataset.

| Algorithm | STL-10-LT | | | | SVHN-LT | | | |
|---|---|---|---|---|---|---|---|---|
| | $N_1 = 150, M_1 = 100k$ | | $N_1 = 450, M_1 = 100k$ | | $N_1 = 1500, M_1 = 3000$ | | $N_1 = 1500, M_c = 3000$ | |
| | $\gamma_l = 20$ $\gamma_u = N/A$ | $\gamma_l = 15$ $\gamma_u = N/A$ | $\gamma_l = 20$ $\gamma_u = N/A$ | $\gamma_l = 15$ $\gamma_u = N/A$ | $\gamma_l = 200$ $\gamma_u = 200$ | $\gamma_l = 150$ $\gamma_u = 150$ | $\gamma_l = 150$ $\gamma_u = 1$ | $\gamma_l = 150$ $\gamma_u = 1/150$ |
| Supervised | 40.44 ±1.42 | 42.31 ±0.42 | 56.56 ±1.07 | 59.81 ±0.45 | 84.10 ±0.05 | 86.14 ±0.50 | 86.14 ±0.50 | 86.14 ±0.50 |
| FreeMatch (ICLR 23) | 70.67 ±0.83 | 70.58 ±0.17 | 76.66 ±0.32 | 77.40 ±0.31 | 90.87 ±1.01 | 91.66 ±0.21 | 94.66 ±0.41 | 88.01 ±0.87 |
| w / SAW (ICML 22) | 71.27 ±0.69 | 70.91 ±0.54 | 78.07 ±0.06 | 78.15 ±0.44 | 89.04 ±0.59 | 90.09 ±0.16 | 94.57 ±0.24 | 89.22 ±1.18 |
| w / Adsh (ICML 22) | 67.37 ±1.15 | 68.10 ±0.10 | 73.44 ±0.70 | 74.43 ±0.14 | 90.22 ±0.45 | 91.78 ±0.22 | 94.21 ±0.57 | 88.47 ±0.66 |
| w / DePL (CVPR 22) | 70.89 ±0.40 | 70.47 ±0.48 | 77.41 ±0.11 | 77.47 ±0.47 | 90.64 ±0.61 | 91.44 ±0.37 | 94.62 ±0.32 | 87.74 ±1.06 |
| w / ACR (CVPR 23) | 69.89 ±0.54 | 70.98 ±0.57 | 78.51 ±0.26 | 79.35 ±0.35 | 84.78 ±1.11 | 87.00 ±0.75 | 92.89 ±0.35 | 91.21 ±0.09 |
| w / BaCon (AAAI 24) | 71.51 ±0.22 | **71.69** ±0.69 | 78.93 ±0.13 | 79.34 ±0.72 | 91.13 ±1.10 | 92.45 ±0.18 | 94.82 ±0.24 | 92.85 ±0.44 |
| w / CPE (AAAI 24) | 69.73 ±0.25 | 70.46 ±0.54 | 78.84 ±0.13 | 79.18 ±0.54 | 91.83 ±0.35 | 92.21 ±0.03 | 94.32 ±0.41 | 90.39 ±0.41 |
| w / SimPro (ICML 24) | 43.30 ±3.18 | 46.05 ±1.22 | 66.70 ±1.69 | 68.03 ±1.17 | 93.17 ±0.35 | **93.73** ±0.06 | **94.86** ±0.22 | 93.95 ±0.29 |
| w / Meta-Expert (Ours) | **71.57** ±0.40 | 71.60 ±1.00 | **78.94** ±0.08 | **79.40** ±0.25 | **93.21** ±0.09 | 93.67 ±0.07 | 94.80 ±0.37 | **94.50** ±0.33 |

# D. Proof of Theorem 1

We first copy the Theorem 1 here.

**Theorem 1.** *Suppose that the loss function $\ell(f(x), y)$ is $\rho$-Lipschitz with respect to $f(x)$ for all $y \in \{1, \ldots, C\}$ and upper-bounded by $U$. Given the class membership $\eta \in \{1, \ldots, Q\}$ and overall pseudo-labeling error $\epsilon > 0$, if $\frac{1}{M} \sum_{j=1}^{M} \sum_{k=1}^{Q} \mathbb{I}(\eta_{j,k} = 1) |\mathbb{I}(max(f_k(x_j)) > t) - \mathbb{I}(\hat{y}_j = y_j)| \leq \epsilon$, for any $\delta > 0$, with probability at least $1 - \delta$, we have:*

$$R(\hat{f}) - R(f^*) \leq 2U\epsilon + 4\sqrt{2}\rho \sum_{y=1}^{C} \mathcal{R}_O(\mathcal{H}_y) + 2U\sqrt{\frac{\log \frac{2}{\delta}}{2O}}. \tag{8}$$

*Proof.* We first derive the uniform deviation bound between $R(f)$ and $\widehat{R}(f)$ by the following lemma.

**Lemma 1.** *Suppose that the loss function $\ell(f(x), y)$ is $\rho$-Lipschitz with respect to $f(x)$ for all $y \in \{1, \ldots, C\}$ and upper-bounded by $U$. For any $\delta > 0$, with probability at least $1 - \delta$, we have:*

$$|R(f) - \widehat{R}(f)| \leq 2\sqrt{2}\rho \sum_{y=1}^{C} \mathcal{R}_{N+M}(\mathcal{H}_y) + U\sqrt{\frac{\log \frac{2}{\delta}}{2(N+M)}}, \tag{9}$$

*where the function space $\mathcal{H}_y$ for the label $y \in \{1, \ldots, C\}$ is $\{h : x \longmapsto f_y(x) | f \in \mathcal{F}\}$.*

*Proof.* In order to prove this lemma, we define the Rademacher complexity of the composition of loss function $\ell$ and model $f \in \mathcal{F}$ with $N$ labeled and $M$ unlabeled training samples as follows:

$$\mathcal{R}_{N+M}(\ell \circ \mathcal{F})$$
$$= \mathbb{E}_{(x,y,\mu)} \left[ \sup_{f \in \mathcal{F}} \sum_{i=1}^{N} \mu_i \Big( \ell\big(f(x_i), y_i\big) \Big) + \sum_{j=1}^{M} \mu_j \Big( \ell\big(f(x_j), y_j\big) \Big) \right]$$
$$\leq \sqrt{2}\rho \sum_{y=1}^{C} \mathcal{R}_{N+M}(\mathcal{H}_y), \tag{10}$$

where $\circ$ denotes the function composition operator, $\mathbb{E}_{(x,y,\mu)}$ denotes the expectation over $x$, $y$, and $\mu$, $\mu$ denotes the Rademacher variable, $\sup_{f \in \mathcal{F}}$ denotes the supremum (or least upper bound) over the function set $\mathcal{F}$ of model $f$. The second line holds because of the Rademacher vector contraction inequality (Maurer, 2016).

Then, we proceed with the proof by showing that the one direction $\sup_{f \in \mathcal{F}} R(f) - \widehat{R}(f)$ is bounded with probability at least $1 - \delta/2$, and the other direction $\sup_{f \in \mathcal{F}} \widehat{R}(f) - R(f)$ can be proved similarly. Note that replacing a sample $(x_i, y_i)$ leads to a change of $\sup_{f \in \mathcal{F}} R(f) - \widehat{R}(f)$ at most $\frac{U}{N+M}$ due to the fact that $\ell$ is bounded by $U$. According to the McDiarmid's inequality (Mohri et al., 2018), for any $\delta > 0$, with probability at least $1 - \delta/2$, we have:

$$\sup_{f \in \mathcal{F}} R(f) - \widehat{R}(f) \leq \mathbb{E} \left[ \sup_{f \in \mathcal{F}} R(f) - \widehat{R}(f) \right] + U\sqrt{\frac{\log \frac{2}{\delta}}{2(N+M)}}. \tag{11}$$

According to the result in (Mohri et al., 2018) that shows $\mathbb{E} \left[ \sup_{f \in \mathcal{F}} R(f) - \widehat{R}(f) \right] \leq 2\mathcal{R}_{N+M}(\mathcal{F})$, and further considering the other direction $\sup_{f \in \mathcal{F}} \widehat{R}(f) - R(f)$, with probability at least $1 - \delta$, we have:

$$\sup_{f \in \mathcal{F}} |R(f) - \widehat{R}(f)| \leq 2\sqrt{2}\rho \sum_{y=1}^{C} \mathcal{R}_{N+M}(\mathcal{H}_y) + U\sqrt{\frac{\log \frac{2}{\delta}}{2(N+M)}}, \tag{12}$$

which completes the proof.

$\square$

Then, we can bound the difference between $\widehat{R}_u(f)$ and $\widehat{R}'_u(f)$ as follows.

**Lemma 2.** *Suppose that the loss function $\ell(f(x), y)$ is $\rho$-Lipschitz with respect to $f(x)$ for all $y \in \{1, \ldots, C\}$ and upper-bounded by $U$. Given the class membership $\eta \in \{1, \ldots, Q\}$ and overall pseudo-labeling error $\epsilon > 0$, if $\frac{1}{M} \sum_{j=1}^{M} \sum_{k=1}^{Q} \mathbb{I}(\eta_{j,k} = 1)|\mathbb{I}(max(f_k(x_j)) > t) - \mathbb{I}(\hat{y}_j = y_j)| \leq \epsilon$, we have:*

$$|\widehat{R}'_u(f) - \widehat{R}_u(f)| \leq U\epsilon. \tag{13}$$

*Proof.* Let's first expand $\widehat{R}'_u(f)$:

$$\widehat{R}'_u(f) = \frac{1}{\hat{M}} \sum_{j=1}^{\hat{M}} \sum_{k=1}^{Q} \mathbb{I}(\eta_{j,k} = 1)\mathbb{I}(max(f_k(x_j)) > t)\ell(f_k(x_j), \hat{y}_j). \tag{14}$$

Then, we show its upper bound:

$$\widehat{R}'_u(f) \leq \frac{1}{M} \sum_{j=1}^{M} \sum_{k=1}^{Q} \mathbb{I}(\eta_{j,k} = 1)\ell(f_k(x_j), y_j) + \mathbb{I}(\eta_{j,k} = 1)\mathbb{I}(\hat{y}_j \neq y_j, max(f_k(x_j)) > t)\ell(f_k(x_j), \hat{y}_j)$$

$$\leq \frac{1}{M} \sum_{j=1}^{M} \sum_{k=1}^{Q} \mathbb{I}(\eta_{j,k} = 1)\Big(\ell(f_k(x_j), y_j) + \epsilon_{k,k}\ell(f_k(x_j), \hat{y}_j)\Big)$$

$$\leq \widehat{R}_u(f) + U\frac{1}{Q} \sum_{k=1}^{Q} \epsilon_{k,k} = \widehat{R}_u(f) + U\epsilon, \tag{15}$$

and its lower bound:

$$\widehat{R}'_u(f) \geq \frac{1}{M} \sum_{j=1}^{M} \sum_{k=1}^{Q} \mathbb{I}(\eta_{j,k} = 1)\ell(f_k(x_j), y_j) - \mathbb{I}(\eta_{j,k} = 1)\mathbb{I}(max(f_k(x_j)) \leq t)\ell(f_k(x_j), \hat{y}_j)$$

$$\geq \frac{1}{M} \sum_{j=1}^{M} \sum_{k=1}^{Q} \mathbb{I}(\eta_{j,k} = 1)\Big(\ell(f_k(x_j), y_j) - \epsilon_{k,k}\ell(f_k(x_j), \hat{y}_j)\Big)$$

$$\geq \widehat{R}_u(f) - U\frac{1}{Q} \sum_{k=1}^{Q} \epsilon_{k,k} = \widehat{R}_u(f) - U\epsilon. \tag{16}$$

By combining these two bounds, we can obtain the following result:

$$|\widehat{R}'_u(f) - \widehat{R}_u(f)| \leq U\epsilon, \tag{17}$$

which concludes the proof.

$\square$

Based on the above lemmas, for any $\delta > 0$, with probability at least $1 - \delta$, we have:

$$R(\hat{f}) \leq \widehat{R}(\hat{f}) + 2\sqrt{2}\rho \sum_{y=1}^{C} \mathcal{R}_O(\mathcal{H}_y) + U\sqrt{\frac{\log \frac{2}{\delta}}{2O}}$$

$$\leq \widehat{R}_l(\hat{f}) + \widehat{R}_u(\hat{f}) + 2\sqrt{2}\rho \sum_{y=1}^{C} \mathcal{R}_O(\mathcal{H}_y) + U\sqrt{\frac{\log \frac{2}{\delta}}{2O}}$$

$$\leq \widehat{R}_l(\hat{f}) + \widehat{R}'_u(\hat{f}) + U\epsilon + 2\sqrt{2}\rho \sum_{y=1}^{C} \mathcal{R}_O(\mathcal{H}_y) + U\sqrt{\frac{\log \frac{2}{\delta}}{2O}}$$

$$\leq \widehat{R}_l(f) + \widehat{R}'_u(f) + U\epsilon + 2\sqrt{2}\rho \sum_{y=1}^{C} \mathcal{R}_O(\mathcal{H}_y) + U\sqrt{\frac{\log \frac{2}{\delta}}{2O}} \qquad (18)$$

$$\leq \widehat{R}_l(f) + \widehat{R}_u(f) + 2U\epsilon + 2\sqrt{2}\rho \sum_{y=1}^{C} \mathcal{R}_O(\mathcal{H}_y) + U\sqrt{\frac{\log \frac{2}{\delta}}{2O}}$$

$$\leq \widehat{R}(f) + 2U\epsilon + 2\sqrt{2}\rho \sum_{y=1}^{C} \mathcal{R}_O(\mathcal{H}_y) + U\sqrt{\frac{\log \frac{2}{\delta}}{2O}}$$

$$\leq R(f) + 2U\epsilon + 4\sqrt{2}\rho \sum_{y=1}^{C} \mathcal{R}_O(\mathcal{H}_y) + 2U\sqrt{\frac{\log \frac{2}{\delta}}{2O}},$$

where the first and seventh lines are based on Lemma 1, and three and fifth lines are due to Lemma 2. The fourth line is by the definition of $\hat{f}$. At this point, we have proven the Theorem 1.

$\square$

