# OpenReview forum: "A Square Peg in a Square Hole: Meta-Expert for Long-Tailed Semi-Supervised Learning"
_ICML.cc/2025/Conference — ICML 2025 poster_

### Official Review · Reviewer_LauN · 2025-03-10

**Overall Recommendation:** 4

**Summary:**

This paper addresses the challenge of long-tailed semi-supervised learning (SSL) by proposing a framework that automatically integrates multiple expert knowledge to generate high-quality pseudo-labels, thereby improving SSL performance in imbalanced data settings. The authors analyze three types of long-tailed distribution scenarios — consistent, uniform, and inverse data distributions — to simulate different real-world imbalance conditions. Leveraging the observation that different layers of features capture different levels of semantic information, the paper introduces a multi-layer feature fusion (DEA loss) mechanism that enables the aggregation of rich and diverse representations for better learning. The proposed loss combines three components: (1) a base SSL loss for standard semi-supervised objectives, (2) a meta-learning loss to aggregate the expertise of multiple models, and (3) the DEA loss to fuse multi-depth features. Additionally, a theoretical analysis is presented to establish a generalization error bound, which supports the robustness of the framework. Extensive quantitative and qualitative experiments on standard benchmarks demonstrate the superiority of the proposed approach over existing baselines.

**Claims And Evidence:**

Most of the claims are provided with evidence.

**Essential References Not Discussed:**

Missing several references:
- Huang et al., FlatMatch: Bridging Labeled Data and Unlabeled Data with Cross-Sharpness for Semi-Supervised Learning, in NeurIPS 2023.
- Yang et al., Robust semi-supervised learning by wisely leveraging open-set data, in TPAMI 2024.
- Lee et al., (FL)2: Overcoming Few Labels in Federated Semi-Supervised Learning, in NeurIPS 2024.

**Experimental Designs Or Analyses:**

Yes, the experimental design is reasonable.

**Methods And Evaluation Criteria:**

N/A

**Other Comments Or Suggestions:**

N/A

**Other Strengths And Weaknesses:**

Strengths:
- Clear and Well-Structured Presentation:
  - The paper is well-written, logically organized, and easy to follow. The motivations, methodology, and results are clearly articulated, making it accessible to a broad research audience.
- Comprehensive Experimental Evaluation:
  - The experimental results show substantial improvements over a wide range of strong baseline methods, indicating that the proposed framework is effective and competitive.
  - The inclusion of both quantitative and qualitative analysis enriches the experimental section, providing deeper insights into the behavior of the proposed method.
- Theoretical Justification:
  - The inclusion of a generalization error bound lends theoretical credibility to the method, which is often lacking in many applied SSL works. This analysis strengthens the contribution and demonstrates a solid understanding of the underlying learning dynamics.

Weaknesses and Concerns:
- Unclear Real-World Relevance of Distribution Scenarios:
  - Although the paper defines three types of data distributions (consistent, uniform, inverse), their practical meaning and relevance to real-world scenarios are insufficiently justified.
  - These settings appear to be artificial constructs, and while they provide controlled experimental environments, it is unclear how frequently such distributions occur in real applications.
  - It would greatly improve the paper to discuss realistic cases where such distributions might naturally arise (e.g., medical diagnosis, fraud detection), or to propose alternative data splits grounded in real-world statistics.
- Lack of Intuition and Justification for Framework Design:
  - Although the overall methodology — aggregating multiple experts and multi-layer feature fusion — is intuitively reasonable, the specific architectural choices are not well-justified.
  - The combination of multiple MLP-based experts and fusion mechanisms seems ad hoc. It remains unclear why these components are structured this way and what alternative designs were considered.
  - More insight into the design rationale, possibly supported by ablation studies, would clarify why this particular form of integration is necessary or optimal.
- Training Stability and Optimization Concerns:
  - Given the complex interplay between multiple expert modules and feature fusion components, optimization stability becomes a natural concern.
  - However, the paper does not provide a discussion on training stability, nor does it offer empirical evidence (e.g., loss curves, variance across runs) to assure readers of its robustness during training.
  - Addressing how gradient conflicts, convergence difficulties, or sensitivity to hyperparameters are handled would greatly improve the trustworthiness of the method.
- Computational Complexity and Efficiency Not Addressed:
  - The proposed method introduces numerous additional components, including multiple expert models and cross-depth feature fusion layers, which likely result in significant computational overhead.
  - Yet, no analysis or discussion on computational cost (e.g., training time, memory usage, inference latency) is provided, which is critical for assessing the practical deployability of the method.
  - Providing comparisons of resource consumption with baseline methods would offer a more balanced view of the method's cost-benefit trade-off.

**Questions For Authors:**

Please check the weaknesses.

**Relation To Broader Scientific Literature:**

This paper enhances the LTSSL with a novel training strategy which could be potentially impactful.

**Theoretical Claims:**

I didn't check the proof carefully.

---

> ### Author Rebuttal · Authors · 2025-04-01
>
> #### **References:**
> We appreciate the reviewer's suggestion and confirm that the highlighted literature is relevant to our study, and we will include citations in the revised manuscript.
> #### **Realistic cases (Weaknesses 1):**
> In the medical field, when collecting information from various patients, we may obtain a long-tailed dataset from non-specialized hospitals, i.e., a large number of common disease cases (head classes) accompanied by very few rare disease cases (tail classes). However, if we consider specialized hospitals focused on specific rare diseases, this scenario would yield an **inverse long-tailed dataset** characterized by abundant rare disease cases and scarce common disease cases. We will add this example in the final version.
>
> In practical applications, where the **distribution of unlabeled data is unknown**, we follow recent works to investigate **three representative extreme distribution** scenarios. If a model can perform well on all those three extreme cases like ours, it is expected to fit different unlabeled data distributions.
> #### **Model design (Weaknesses 2):**
> To effectively address three different unlabeled data distributions, we constructed three MLP-based experts with logit adjustment, specifically designed for long-tailed, uniform, and inverse long-tailed distributions, respectively. Furthermore, we observed that different depth features and differnt experts exhibit its own properties (as evidenced in Tables 1-3 and Fig. 1). To fully leverage the characteristics of multi-expert and multi-layer features, we employed DEA and MFF for respective expert and feature integration. Table 6 validates the effectiveness of DEA and MFF. Moreover, we investigated the performance differences between addition-based and concatenation-based feature operation strategies for feature fusion in Table 9, ultimately adopting the superior addition-based feature operation strategy.
> #### **Training Stability (Weakness 3):**
> Our empirical validation demonstrates consistent optimization behavior across multiple runs, as evidenced by the **lower standard deviations** reported in Tables 4-5 and 7-8 (which are comparable to or better than those of baselines). We further provide the accuracy curve in https://anonymous.4open.science/r/Acc-curve/Acc-curve.png, which shows that our method performs stable in the later stages of training.
> #### **Computational Complexity (Weankess 4):**
> Please refer to the response to reviewer wN5j's Computational overhead (Weakness 2).

---

> > ### Comment · Reviewer_LauN · 2025-04-04
> >
> > Thanks for the detailed comments from the authors. My concerns have been addressed through both intuitive justification and empirical evaluations, therefore, I am willing to raise my score accordingly.

---

### Official Review · Reviewer_65rT · 2025-03-12

**Overall Recommendation:** 2

**Summary:**

In response to the problem of distribution mismatch between labeled and unlabeled data in long-tail semi-supervised learning (LTSSL), current methods refer to experts to model various unlabeled data, but various experts cannot match long-tail pseudo-labeled data well. This paper proposes a dynamic expert allocation method to solve the problem of long-tail unlabeled data, while connecting multi-scale feature fusion to improve the classification accuracy of LTSSL. Through a large number of experiments, it has been proven that the method proposed in the paper is superior to the SOTA methods.

**Claims And Evidence:**

Please provide specific examples regarding the mismatch between labeled and unlabeled data distributions in LTSSL.

**Essential References Not Discussed:**

Not found yet.

**Experimental Designs Or Analyses:**

1. Compared with theBacon,  ACR and CPE methods, the experiments in this paper lack comparison on the CIFAR100-LT and ImageNet-127 datasets.
2. When compared with baseline methods such as CPE, the accuracy values of the baseline methods mentioned in Table 4 of this paper were not found in the original text, such as when r_l=200,  r_u=200. I wonder how this paper obtained the accuracy values of various methods for this condition? Similar situations also appear in Table 5, such as STL-10-LT.
3. In the ablation study, as shown in Table 6, when r_u=200, 1, 1/200, the accuracy using DEA module is similar to that without DEA module. Does this indicate that the role of DEA module is not significant?

**Methods And Evaluation Criteria:**

The proposed method in the paper make sense for the problem or application at hand.

**Other Comments Or Suggestions:**

Please refer to the question section.

**Other Strengths And Weaknesses:**

Strengths of the paper:
1. The paper focuses on the attention range of different experts in LTSSL, and the accuracy of each expert's prediction of pseudo labels depends on the category of their respective training data. But the paper does not delve deeply into the issue, and the prediction accuracy of different experts is only related to the number of corresponding categories?
2. The paper is well-organized, clearly expressed, and easy to read and understand.

Weakness of the paper:
1. The first discovery of the characteristics of features at different scales mentioned in the paper is a bit exaggerated. In deep learning, the characteristics of features at different scales have already been mentioned.
2. The DEA module mentioned in the paper lacks a detailed introduction to the scope of expert attention.
3. The experimental part of the paper has unclear data sources and lacks some of the datasets mentioned in the published papers.

**Questions For Authors:**

1. At the beginning of model training, how to solve the parameter training of DEA module under poor pseudo-label conditions?  Because the parameter optimization of DEA module largely relies on labeled data and unlabeled data, and the pseudo-label accuracy of unlabeled data is poor.
2. Does the experimental dataset fully reference the datasets of other long -tail semi-supervised training methods?
3. What are the specific differences between the three expert fusion strategies proposed in this paper and the CPE method?
4. How to quantitatively define the recognition scope of each expert in the fusion of the three experts proposed in this article? What is the recognition range of long tail experts?
5. The advantages and disadvantages of multi-depth features have been discovered by researchers for a long time, and this paper insists on presenting them for the first time, which is a bit exaggerated. The MFF module's contribution is somewhat limited.  The ablation study also showed that the role of the MFF module was not significant.
6. If the predicted labels y_ {m, j} ^ {u} and pseudo-labels \hat {y}_j are obtained through an aggregator, are these two aggregators the same model or different models? How to update the parameters of two models to ensure optimal performance?

**Relation To Broader Scientific Literature:**

The method proposed in the paper is inspired by the published CPE method and improves the shortcomings of experts in CPE.

**Theoretical Claims:**

I have checked the theoretical proof, but I am not quite sure about its correctness.

---

> ### Author Rebuttal · Authors · 2025-04-01
>
> #### **More experiments (Experimental Design 1):**
> Following recent works like CPE and BaCon, we conducted experiments across three datasets (CIFAR-10-LT, STL-10-LT, and SVHN-LT), each evaluated under two different imbalance ratios (150 and 200). As suggested, we have extended our evaluation to include CIFAR-100-LT, along with conducting evaluations on CIFAR-10-LT and STL-10-LT under lower imbalance ratios. The results presented in Tables R2 and R3 [refer to the responce to reviewer sReE's Questions 1 (Aligned experimental setting)] demonstrate that **our method achieves comparable or superior performance even on these standardized benchmarks, with more significant performance improvements observed under higher imbalance ratios.**
> #### **Results of CPE (Experimental Design 2 & Weakness 3 & Question 2):**
> We rigorously evaluated baselines (like CPE) **using their official codes under the specified conditions**, re-implementing experiments with matched hyperparameters and three independent runs.
> #### **Effectivness of DEA and MFF (Experimental Design 3):**
> The ablation results in Table 6 demonstrate that: the **DEA module provides an average 1.68% accuracy improvement** across all imbalance ratios (γ_u = 200/1/1/200), showing relatively greater performance gains; while the **MFF module delivers a 0.76% average gain**, which though comparatively smaller, remains statistically significant. The **combined DEA+MFF configuration achieves 2.42% improvement**, confirming their complementary effectiveness and synergistic interaction. These improvements conclusively validate the effectiveness of our proposed modules in enhancing model robustness across diverse imbalance ratios.
> #### **Multi-scale feature (Weaknesses 1 and Questions 5):**
> We acknowledge existing discussions about multi-scale feature characteristics in deep learning. However, to the best of our knowledge, **no prior work has investigated how long-tailed distributions differentially affect shallow and deep features**. Our analysis reveals that shallow features are relatively balanced although less discriminative, and deep features improve the discriminative ability but are less balanced. We strategically leverage this complementary characteristic to achieve performance gains through fusion of different depth features. Specifically, the ablation results in Table 6 demonstrate that the MFF module **delivers a 0.76% average gain**, confirming its significant effectiveness.
> #### **Expert attention (Weaknesses 2 and Questions 4):**
> The expert attention scope in our DEA module **follows the partitioning in CPE** while maintaining comparable performance to alternative scoping strategies. As shown in Table R5, we conducted additional experiments with alternative splits on CIFAR-10-LT. The average performance fluctuation of our method during partitioning changes is **only 0.37%**, which is smaller than CPE's 0.81%, verifying the DEA module's robustness regardless of specific partitions.
> #### Table R5: **CIFAR-10-LT** with (N_1, M_1, γ_l) = (1500, 3000, 200)
> ||γu=200|γu=1|γu=1/200|
> |-|-|-|-|
> |CPE[2, 2, 6]|78.57|83.47|84.40|
> |CPE[3, 3, 4]|77.72|82.76|83.52|
> |Ours[2, 2, 6]|81.67|83.96|85.75|
> |Ours[3, 3, 4]|80.92|84.07|85.51|
> #### **DEA training (Questions 1):**
> To address parameter optimization for the DEA module under poor initial pseudo-labels, we introduced a confidence threshold (t = 0.95) to filter unreliable pseudo-labels following CPE and ACR. To quantify its impact, we evaluated the pseudo-label accuracy with and without the threshold in the following Table R6, which suggests **incorporating threshold improves accuracy by 3.85% on average**.
> #### Table R6: **CIFAR-10-LT** with (N_1, M_, γ_l) = (1500, 3000, 200)
> ||γu=200|γu=1|γu=1/200|
> |-|-|-|-|
> |Ours without threshold|84.39|81.80|91.65|
> |Ours (including threshold)|**91.71**|**82.87**|**94.81**|
> #### **Fusion strategy (Questions 3):**
> CPE does not use fusion strategy and employs three experts simultaneously to produce pseudo-labels for all samples, along with a uniform expert to make predictions. In contrast, our DEA module **learns class membership automatically** and utilizes it to select the most appropriate single expert for each sample. This strategy avoids error accumulation caused by conflicting predictions in CPE, thereby significantly improving pseudo-label quality. Our ablation study in Table 6 quantitatively demonstrates that DEA achieves an average 1.68% accuracy improvement across all imbalance ratios (γ_u = 200, 1, 1/200) compared to CPE.
> #### **Aggregator (Questions 6):**
> The aggregator model processes both strong and weak augmentation views simultaneously, generating $y_{m, j}^{u}$ and $\hat{y_j}$ respectively. The aggregator’s parameters are updated through a unified optimization process. Specifically, the model is trained to minimize the consistency loss between $y_{m, j}^{u}$ and $\hat{y_j}$, ensuring that predictions align across augmentation views.

---

> > ### Comment · Reviewer_65rT · 2025-04-07
> >
> > The author's feedback did not fully address my issue, so I maintain my original evaluation.

---

> > > ### Author Response · Authors · 2025-04-07
> > >
> > > We appreciate your valuable comments. Due to space constraints in the original response, some details might not have been sufficiently elaborated. We will further address these points comprehensively:
> > > ### **Specific examples of mismatch (Claims And Evidence)**:
> > > In the medical field, when collecting clinical data, we may obtain a long-tailed dataset from hospitals, i.e., many common disease cases (head classes) accompanied by very few rare disease cases (tail classes). However, the clinical  data collected from a wide range of populations is unlabeled and characterized by an abundance of non-diseased individuals and a scarcity of diseased individuals, especially those with rare diseases. Thus, the unlabeled data distribution is **mismatched** with the labeled data distribution. **We will add this example in the final version**.
> > > ### **Theoretical proof(Theoretical Claims)** :
> > > We provide **a generalization error bound** for our method. The key points of the proof and reasoning are as follows:
> > >
> > > At first, we derive the uniform deviation bound between empirical risk $R(f)$ and true risk $\widehat R(f)$ (unlabeled data with ground-truth labels) using Rademacher complexity and McDiarmid's inequality. The conclusion is shown in Eq.(9).
> > >
> > > Then, we bound the difference between $\widehat R_u(f)$ and $\widehat R_u^{\prime}(f)$ (unlabeled data with pseudo-labels) based on the definition of consistency loss. The conclusion is shown in Eq.(13).
> > >
> > > At last, by using our DEA, we make $\epsilon$ consist three parts, and each denotes the pseudo-labeling error of a specific expert on the unlabeled data located in its attention scope. Building upon Eq.(9)(Lemma 1)and Eq.(13)(Lemma 2), we derive the Eq.(18).
> > >
> > > The generalization error bound quantifies how the model's performance on unseen data relates to its performance on the training data. The bound mainly depends on two factors: **the overall pseudo-labeling error ($ϵ$)** and **the number of training samples ($O$)**. As $ϵ \rightarrow 0$ and $O \rightarrow \infty$, the empirical risk minimizer ($\hat f$) converges to the true risk minimizer ($f^∗$). As demonstrated in Table 3, **our method significantly reduces the pseudo-labeling error compared to previous methods**, and thus improves the model's performance.
> > > ### **Experiments(Experimental Design 1&Experimental Design 2&Weakness 3&Question 2)**:
> > > Recent studies almost used 3-4 long-tailed benchmark datasets: CPE was evaluated on CIFAR-10-LT, CIFAR-100-LT, and STL-10-LT, BaCon on CIFAR-10-LT, CIFAR-100-LT, STL-10-LT, and SVHN-LT, and ACR on CIFAR-10-LT, CIFAR-100-LT, STL-10-LT, and ImageNet-127. We also conducted experiments across three datasets (CIFAR-10-LT, STL-10-LT, and SVHN-LT) with **higher imbalance ratios** (the **more challenging scenarios**) compared to recent works. All baselines were rigorously evaluated **using their official codes under specified conditions**, with experiments re-implemented with matched hyperparameters and three independent runs to compensate for unreported accuracy values in the original publications.
> > >
> > > As suggested, we have further extended our evaluation on a new dataset CIFAR-100-LT, along with conducting evaluations on CIFAR-10-LT and STL-10-LT under lower imbalance ratios to align with the previous works. The results presented in Tables R2 and R3[**refer to the responce to reviewer sReE's Questions 1 (Aligned experimental setting)**] demonstrate that our method achieves **comparable or superior performance on these standardized benchmarks**. Moreover, our advantages become even **more pronounced under higher imbalance ratios**: on CIFAR-10-LT with(N1,M1)=(1500,3000), our method **achieves performance gains of +0.23%(γl=100), +1.26%(γl=150), and +1.64%(γl= 200)over previous SOTA methods.**
> > > ### **Expert attention (Strengths 1&Weaknesses 2&Questions 4)**:
> > > We‘d like to clarify three key points:
> > >
> > > First, in the long-tailed distribution, a small portion of classes(head classes) have a massive number of samples, while a large proportion of classes(tail classes) are associated with only a few samples. However, there is **no unified definition for the exact number of head, medium, or tail classes**. The expert attention scope in our DEA module **aligns with the class partitioning established in CPE**.
> > >
> > > Second, in our method, **each expert is train on all training data but with different logit adjustment intensities**, and thus **long tailed/uniform/inverse long tailed expert is skilled in samples located in the head/medium/tail interval**, as evidenced in Table 1.
> > >
> > > Third, in Table R5, we conducted experiments with other splits on CIFAR-10-LT. We can observe that the **average performance fluctuation** of our method during splits changes is only 0.37%, which is **smaller** than CPE's 0.81%; our method **consistently outperformance** CPE with alternative splits.
> > >
> > > In summary, the prediction accuracy of different experts is not related to the number of classes located in its corresponding attention scope.

---

### Official Review · Reviewer_sReE · 2025-03-13

**Overall Recommendation:** 3

**Summary:**

This paper proposes Meta-Expert, a semi-supervised learning method tackling the long-trained problem. By investigating the effectiveness of assiging different experts regarding the class membership, the model applies a dynamic expert assignment model to learn to assign the soft weight for three expert models. The method is futher improved by a feature fusion module to balance the assignment. The experimental results prove the effectiveness of proposed method.

**Claims And Evidence:**

The claims are well supported by the evidence.

**Essential References Not Discussed:**

The related works are referenced.

**Experimental Designs Or Analyses:**

The experimental designs is convincing. However, the data setting is not consist with recent LTSSL works. Please refer to the Question part.

**Methods And Evaluation Criteria:**

The proposed methods and evaluation criteria make sense.

**Other Comments Or Suggestions:**

Please refer to the Question part.

**Other Strengths And Weaknesses:**

Please refer to the Question part.

**Questions For Authors:**

1. The data setting in Table.4 (e.g., the setting of $N_1$, $M_1$, $\gamma_l$, $\gamma_u$) is not consist with recent works [1,2]. More experiments on the aligned setting are expected to make the result convincing.
2. As shown in Table. 1, Meta-Expert achieves a significant higher performance than Upper-E, which use the ground-truth membership to align the experts. Does this result mean that GT membership is not the best alignment target?

[1] Three Heads Are Better Than One: Complementary Experts for Long-Tailed Semi-supervised Learning.

[2] SimPro: A Simple Probabilistic Framework Towards Realistic Long-Tailed Semi-Supervised Learning.

**Relation To Broader Scientific Literature:**

This work is related to the long-tailed semi-supervised learing task. Previous LTSSL methods tackled this problem by various techniques like using the fusion output from multi-head classifier. This work further propose a meta aggregation module to automatically assign the class to experts.

**Theoretical Claims:**

I check the analysis of generalization error bound in Sec 3.4.

---

> ### Author Rebuttal · Authors · 2025-04-01
>
> #### **Questions 1 (Aligned experimental setting):**
> Our work focuses on advancing long-tailed semi-supervised learning, thus primarily investigating settings with higher imbalance ratios (the more challenging scenarios) compared to recent works. As suggested, we have conducted supplementary experiments on CIFAR-10-LT, CIFAR-100-LT, and STL-10-LT under lower imbalance ratios aligned with recent works. As empirically demonstrated in Tables R2 and R3, **our method achieves comparable or superior performance even in these standardized benchmarks**: attaining better results in 5/6 cases and comparable performance in 1/6 cases on CIFAR-10-LT (with maximum gains of +1.19%), showing +1.06% improvement on STL-10-LT and +0.58% average improvement on CIFAR-100-LT. Moreover, **our advantages become even more pronounced under higher imbalance ratios: on CIFAR-10-LT with (N_1, M_1) = (1500, 3000), our method achieves performance gains of +0.23% (γ_l = 100), +1.26% (γ_l = 150), and +1.64% (γ_l = 200) over previous SOTA methods.**
> #### Table R2: **CIFAR-10-LT** with (N1, M1, γl) = (500, 4000, 100) (left three columns) and (N1, M1, γl) = (1500, 3000, 100) (right three columns)
> |             | γu=100 | γu=1 | γu=1/100 | γu=100 | γu=1 | γu=1/100 |
> |-------------|--------|------|----------|--------|------|----------|
> | BaCon       | 80.82 | 79.79 | 79.61 | 83.13 | 82.66 | 85.94 |
> | CPE         | 80.68 | **82.32** | 83.88 | 84.44 | 85.86 | 87.09 |
> | SimPro      | 72.77 | 71.78 | 73.05 | 82.33 | 80.25 | 83.22 |
> | Ours        | **82.01** | 82.01 | **83.94** | **84.94** | **86.13** | **87.32** |
>
> #### Table R3: **STL-10-LT** with (N1, M1, γl) = (150, 100K, 10) (left one column) and **CIFAR-100-LT** with (N1, M1, γl) = (150, 300, 10) (right three columns)
> |STL-10-LT           | γu=N/A |CIFAR-100-LT       | γu=10 | γu=1 | γu=1/10 |
> |-----------|--------|-------|-------|------|---------|
> | BaCon     | 71.15 | BaCon | 60.05 | 50.21 | 60.30 |
> | CPE       | 73.07 | CPE   | 59.83 | 48.09 | 60.83 |
> | SimPro    | 50.91 | SimPro| 59.04 | 48.25 | 60.09 |
> | Ours      | **74.13** | Ours  | **60.21** | **50.73** | **61.88** |
> #### **Questions 2 (Upper-E):**
> **Upper-E denotes CPE directly use GT membership to select a specific expert to produce pseudo-labels and make predictions**, while our method utilizes the integration of the logits from the three experts to compute the loss in Eq. (4), which may push different experts learn better. Moreover, as shown in Eq. (3), the final prediction of our method is the soft ensemble of multiple experts, which further brings better performance.
>
> To provide empirical validation, we replaced our DEA module with the GT membership and conduct experiment on CIFAR-10-LT. The experimental results in Table R4  demonstrate that: i) **The design motivation for employing the DEA module to orchestrate expert collaborations through learned membership relationships is empirically well-founded**, ii) The MFF module can helps experts toward more optimal direction by fusing different depth features.
> #### Table R4: **CIFAR-10-LT** with (N_1, M_1, γ_l) = (1500, 3000, 200)
> |        | γu=200 | γu=1 | γu=1/200 |
> |--------|------|--------|----------|
> | CPE    | 78.57 | 83.47 | 84.40 |
> | **Upper-E** (CPE + GT membership) | 79.86 | 86.14 | 88.03 |
> | Ours   | 81.67 | 83.96 | 85.75 |
> | **Upper-E+** (Ours + GT membership) | **85.04** | **86.51** | **88.60** |

---

> > ### Comment · Reviewer_sReE · 2025-04-02
> >
> > The author has addressed my questions. Therefore, I would like to raise my score.

---

### Official Review · Reviewer_wN5j · 2025-03-14

**Overall Recommendation:** 4

**Summary:**

This paper introduces Meta-Expert, a framework designed for long-tailed semi-supervised learning. Specifically, Meta-Expert includes a Dynamic Expert Assignment module, which predicts the class membership of a sample. A Multi-Depth Feature Fusion module, which integrates features from different depths to mitigate bias. Through extensive experiments, Meta-Expert achieves new state-of-the-art performance. Additionally, the authors conduct ablation studies to analyze the importance of different modules.

**Claims And Evidence:**

N/A

**Essential References Not Discussed:**

Although the authors discuss many related works, they still miss some highly relevant studies, such as:
* BMB: Balanced Memory Bank for Long-Tailed Semi-Supervised Learning, TMM 2024
* CoSSL: Co-Learning of Representation and Classifier for Imbalanced Semi-Supervised Learning, CVPR 2022

**Experimental Designs Or Analyses:**

The experiment is relatively extensive.

**Methods And Evaluation Criteria:**

The proposed method Meta-Expert makes sense for the LTSSL problem.

**Other Comments Or Suggestions:**

N/A

**Other Strengths And Weaknesses:**

Strengths:
* This paper is well-written and well-motivated.
* The MFF and DEA modules proposed in this paper are effective and relatively novel.

Weaknesses:
* The paper does not analyze the reasons behind the different properties (such as separability and bias) of features from different layers.
* Due to the introduction of new modules, such as the three experts, MFF, and DEA modules, an analysis of the additional computational overhead is needed.

**Questions For Authors:**

Which specific layers do the shallow, medium, and deep layers correspond to? Does the choice of different layers have a significant impact?

**Relation To Broader Scientific Literature:**

Meta-Expert reveals that features from different layers exhibit distinct characteristics in terms of separability and bias distribution, which could be beneficial for future research.

**Theoretical Claims:**

N/A

---

> ### Author Rebuttal · Authors · 2025-04-01
>
> #### **References:**
> We appreciate the reviewer's suggestion and confirm that the highlighted literature is relevant to our study, and we will include citations in the revised manuscript.
> #### **Properties of different layers (Question & Weakness1):**
> In deep networks, shallow layers capture local patterns while deep layers learn global semantics. For long-tailed learning, since head and tail classes may share similar local patterns, shallow features exhibit balanced discriminability across classes. Meanwhile, deep layers predominantly encode head class semantics due to their overwhelming sample dominance, thus biasing predictions toward head classes. This is empirically supported by Table 2: **shallow features are relatively balanced although less discriminative, and deep features improve the discriminative ability but are less balanced**.
>
> Our backbone network employs WideResNet-28-2, which consists of three convolutional block groups. We use the outputs of these three block groups (the 10th, 19th, and 28th layers) to represent shallow, middle, and deep features respectively. While our primary analysis focuses on features from these three specified layers, the phenomenon observed in Table 2 is general applicability. To verify this, we conducted extended experiments using alternative intermediate layers (the 6th, 15th, and 24th layers), with supplementary results presented in the floowing table, which aligns with Table 2's observations, suggesting that the selection of different layers does not significantly impact the revealed observations.
> #### Table R1
> | Feature depth | Overall | Head   | Medium | Tail   | GAP    |
> |---------------|---------|--------|--------|--------|--------|
> | 6th layer       | 24.09   | 23.77  | 29.23  | 17.57  | 6.20   |
> | 15th layer       | 35.16   | 39.77  | 37.60  | 27.30  | 12.47  |
> | 24th layer       | 44.61   | 59.10  | 41.15  | 34.73  | 24.37  |
> #### **Computational overhead (Weakness 2):**
> While our proposed modules (three experts, MFF, and DEA) introduce a controlled parameter increase of 13.3% (1.5M → 1.7M), this design achieves strategically balanced efficiency-performance trade-offs. Experimental results on CIFAR-10-LT demonstrate: a 6.4% increase in epoch time (234.5s → 249.5s), a 1.6s increased in inference time for evaluating 10,000 samples (7.1s → 8.8s), and a substantial accuracy gain of +3.5% absolute improvement (71.9% → 74.4%), collectively indicating significant performance enhancement with modest computational overhead.

---

### Decision · Program_Chairs · 2025-05-01

**Decision:**

Accept (poster)

**Comment:**

The paper received three positive reviews and one negative review.  The strengths of the work include: (relatively) novel idea, effective algorithm, comprehensive evaluation, and some theoretical analysis. There are a few weaknesses, such as insufficient analysis of the property, lack of optimization stability, and computation cost, which, however, have been addressed by the rebuttal.

The negative rating was given by Reviewer 65rT. The weaknesses pointed out by the reviewer are:  1) The first discovery of the characteristics of features at different scales mentioned in the paper is a bit exaggerated; 2) The DEA module mentioned in the paper lacks a detailed introduction to the scope of expert attention; 3) The experimental part of the paper has unclear data sources and lacks some of the datasets mentioned in the published papers. Regarding these weaknesses and the concerns of the reviewer, the authors provided a lot of clarification. The reviewer didn't think the rebuttal addressed the concerns.

As far as I am concerned, the issues mentioned by Reviewer 65rT are relatively minor and at least partial of the concerns have been addressed.